# Determinants of adolescents' depression, anxiety, and somatic symptoms in Northwest Ethiopia: A non-recursive structural equation modeling

Zenebe Abebe Gebreegziabher[1]*, Rediet Eristu[2], Ayenew Molla[2]

1 Department of Epidemiology and Biostatistics, School of Public Health, Asrat Woldeyes Health Science Campus, Debre Berhan University, Debre Berhan, Ethiopia, 2 Department of Epidemiology and Biostatistics, Institute of Public Health, College of Medicine and Health Sciences, University of Gondar, Gondar, Ethiopia

* zenebeabe7054@gmail.com

## Abstract

### Introduction

In low and middle-income countries, adolescent mental health is not only a major public health challenge but also a development concern. Depression and anxiety are the most common mental health disorders and somatic symptoms often co-exist with them. Adolescents with common mental health problems are associated with an increased risk of suicide, future unemployment, and poor quality of life. However, little is known about the mental health of adolescents in Ethiopia. Thus, this study aimed to assess the determinants of depression, anxiety, and somatic symptoms among adolescents in Northwest Ethiopia, in 2022.

### Methods

An institution-based cross-sectional study was conducted from June 8 to 24, 2022. Two-stage stratified random sampling was used to select 1407 adolescents in Northwest Ethiopia. Structured and standardized self-administered questionnaires were used to collect the data. Non-recursive structural equation modeling was employed to assess the direct, indirect, and total effects of predictors. Adjusted regression coefficients and corresponding 95% confidence intervals were used to interpret the strength of the association.

### Results

The prevalence of depression, anxiety, and somatic symptoms were 28.21% (95% CI: 25.8, 31%), 25.05% (95%CI: 22.8, 27.5), and 25.24(95% CI: 23, 27.6%) respectively. Alcohol use had a significant positive effect on depression [β = 0.14, 95% CI: 0.073, 0.201], anxiety [β = 0.11, 95% CI: 0.041, 0.188], and somatic symptoms [β = 0.12, 95% CI: 0.062, 0.211]. Stress had a significant positive effect on depression [β = 0.76, 95% CI: 0.642, 0.900], anxiety [β = 1.10, 95% CI: 0.955, 1.264], and somatic symptoms [β = 086, 95% C: 0.700, 1.025]. Depression had a direct positive effect on anxiety [β = 0.74, 95% CI: 0.508, 1.010].

**Data Availability Statement:** All relevant data are within the paper and its Supporting Information files.

**Funding:** The author(s) received no specific funding for this work.

**Competing interests:** The authors have declared that no competing interests exist.

**Abbreviations:** AIC, Akaike information Criteria; OVB, Omitted variable bias; CFI, Comparative Fix index; CMB, Common Method Bias; OVB, Omitted Variable Bias; PHQ, 9A: Patent Health Questionnaire for Adolescent; RMSEA, Rooted Mean Square Error of Approximation; TLI, Tucker Lewis Index,.

## Conclusion

In this study, the prevalence of depression, anxiety, and somatic symptoms was moderate. Alcohol use and stress were significantly related to depression, anxiety, and somatic symptoms. The bidirectional relationship between anxiety and depression was significant. Therefore, public health interventions should focus on the bidirectional relationship between depression and anxiety, as well as on identified factors to reduce the burden of mental illness in adolescents.

## Introduction

Mental health disorders account for 16% of the global burden of disease and injuries and more than 80% of people with mental health disorders live in low and middle-income countries, which is associated with an increase in incidence and mortality [1].

Adolescence marks the transition from infancy to adulthood, involving significant development in physical, psychological, social, and cognitive aspects, which has been linked to confusion, stress, and emotional instability [2]. Adolescents are especially vulnerable to developing mental illness [3]. In this populationmental health disorders increased by 32.18% from 1999 to 2019 [4]. According to the World Health Organization, mental health issues impact 10–20% of adolescents in the world [5]. In low and middle-income countries, adolescent mental health is not only a major public health challenge but also a development concern [6].

Human suffering and financial costs associated with mental health disorders are substantial and growing globally [7]. More than 12 billion working days are lost annually as a result of mental illness. More than diabetes, cancer, and respiratory diseases combined, it is predicted that between 2011 and 2030, the world economy will lose 16 trillion US dollars due to mental illness [6]. Depression and anxiety are among the most common mental health disorders [8], and often somatic symptoms coexist with them [9,10].

Anxiety is characterized by emotional feelings of excessive fear, nervousness, avoiding threats in the environment perceived by them, and physical symptoms such as fast respiration, increased blood pressure, and tightness of the chest [11,12]. Globally, 301.39 million people are affected by anxiety [13]. It is the leading cause of mental disorders in the world, accounting for approximately 28.68 million disability-adjusted life years in the global burden of disease [14]. Anxiety is the most common mental health disorder in adolescents [15], with a high prevalence in developed as well as in developing countries [16]. In adolescents, it ranges from 42.1% [17] to 80.85% in Asia [18], and in sub-Saharan Africa, 29.8% of adolescents had anxiety [3]. A study conducted in Kenya showed that 37.99% of high school adolescents had anxiety [19]. A systematic review conducted among children and youth in Ethiopia revealed that general anxiety disorder ranges from 0.5–23% [20]. In combination with depression, anxiety contributes to 45% of the overall burden of disease [21]. Different pieces of literature reported that female sex [18,22], age [17,19], smoking [23,24], alcohol use, low self-rated academic ability [17,25], family academic pressure [26], stress [27,28], somatic symptom disorder [29] and depression [30] have a positive relationship with anxiety.

Depression, which is the second most common mental health disorder in adolescents, is characterized by loss of interest or pleasure, feelings of guilt or low self-worth, feelings of tiredness, disturbed sleep or appetite, and poor concentration and sadness [31]. In 2018, over one million adolescents died from preventable causes, with depression playing a significant role [32]. In Asia, its prevalence ranges from 24.3% to 57.7% [33], and in Sub-Saharan Africa, it

ranges from 15.5% [34] to 45.90% [19,35]. In Ethiopia, 28% of adolescents in Jimma [36] and 36.2% of adolescents in Aksum [37] were depressed. Previous studies documented that age [19], female sex [36,38–40], being in public school [41], high family academic pressure [26,42], poor self-rated academic ability [43,44], alcohol use [45,46], cigarette smoking [24,47], stress [48–50], somatic symptoms [36,38,39,48,51,52], and anxiety [53,54] had a positive relation with depression. Other factors, such as social support [43,44,51,55], and family education [52] had, negative associations with depression.

A somatic symptom is characterized by a comprehensive list of symptoms such as pain, breathlessness, numbness, palpitation, tiredness, headache, dizziness, and gastroenterological problems. Somatic symptom disorder is a disorder in which individuals excessively or disproportionally think or experience feelings about the symptoms, which results in significant disturbance in daily life [56]. Depression and anxiety often coexist with somatic symptoms [57–59], widespread issues in primary health care and subspecialty settings [10]. Somatic symptoms result in more than half of the patient visits to primary health care [60]. Its prevalence is estimated to be 5 to 7% [61] in the general population and 5 to 30% [62,63] in adolescents. Somatic symptoms delay the diagnosis of depression and anxiety because most people seek medical attention for bodily symptoms. Finally, individuals with somatic symptoms may experience physical harm from unnecessary medical procedures [64]. Different scholars reported that female [65–67], being in private school [65], physical inactivity [68,69], extracurricular tutoring [70], stress [71–73], anxiety [66,74,75] and depression [66,76,77] had significant positive relations with somatic symptoms.

Mental health is a key part of sustainable development goals that affect every other goal [6], yet the prevalence of mental health disorders is still high [18]. To decrease the burden of mental illness, targeting adolescents is important; since more than half of mental health problems start at this age [6], and 50% of adolescent mental health problems will proceed to adulthood [78]. Although adolescents in low and middle-income countries are disproportionally affected by mental health disorders, there is limited data on the prevalence and determinants of mental illness in Sub-Saharan Africa [79,80].

In Ethiopia, although mental health services were included in the national health policy, there is a paucity of evidence on the prevalence and determinants of anxiety, somatic symptoms, and their relation with depression. Adolescents in northwest Ethiopia are especially vulnerable to mental illness, due to repeated internal conflicts in northern Ethiopia [81]. Moreover, studies conducted outside of Ethiopia had methodological flaws, as they used logistic or linear regression to assess multiple mental health outcomes, which cannot address the bidirectional relationship between outcome variables and the indirect effect of predictors. In such cases, non-recursive structural equation modeling, which is a multivariate statistical framework used to measure the complex relationships between different observed and latent variables simultaneously is preferable.

Therefore, this study aimed to assess the prevalence, and determinants of depression, anxiety, and somatic symptoms and the relationship between these variables among high school and preparatory school adolescents in Gondar town by using non-recursive structural equation modeling. The findings from this study will help psychologists, psychiatrists, policymakers, students, the community, and the government to tackle the increasing burden of mental health problems. Moreover, it will be used as input for sustainable development and as baseline data for further research.

## Methods and materials

### Study design and context

An institution-based cross-sectional study was conducted from June 8 to June 24, 2022, among high and preparatory school adolescents in Gondar town, Northwest, Ethiopia. It is

located 728 kilometers from Ethiopia's capital city of Addis Ababa, and 180 kilometers from Bahir Dar, the capital city of the Amhara regional state. There are 395, 000 residents [82], and 48 healthcare facilities including one comprehensive specialized hospital, eight health centers, one private general hospital, fifteen specialist clinics, fifteen medium clinics, and eight primary clinics. Adolescents comprise more than 25% of the Ethiopian population and the health of adolescents determines the future development of the country. According to the town administrative educational department report, the town has seventeen (twelve public and five private high schools), with a total of 24,308 students. Among them, 2408 students were from private schools (1457 girls and 951 boys) and 21900 students were from public schools (12052 girls and 9848 boys).

## Participants

All high and preparatory school adolescents in Gondar town who were registered for the second semester of the 2021–2022 academic years were included in the study. Night preparatory and high school students and students who were transferred in from other areas after January 8, 2022, were excluded.

The sample size was determined using the general rule of thumb approach for calculating sample size in structural equation modeling. This approach suggests a range of 5 to 20 times the number of free parameters [83]. In this study, a 5:1 ratio was used to obtain the minimum adequate sample size. Based on the hypothesized model *Fig 1*, a total of 134 free parameters were available (*Fig 1*).

Considering 134 free parameters, the sample size was

134*5 = 670: Since the sampling procedure is multistage sampling with 2 stages, a design effect of 2 was considered.

Therefore, the sample size becomes 670*2 = 1340

After adjusting for 5% for non-response rate, the final sample size became

1340+1340*5% = **1407**

Therefore, the sample size becomes 670*2 = 1340

After adjusting for 5% for non-response rate, the final sample size became

1340+1340*5% = 1407

To select 1407 participants, a two-stage stratified random sampling technique was employed (*Fig 2*). Out of 1407 randomly selected individuals, 1379 completed the questionnaire; hence, the response rate was 98%. The mean (± SD) age of the respondents was 17.23 ± 1.25 years with a range of 15 to 19 years. The majority, 1238(89.78%) of the respondents were from public schools, and regarding residence, two-thirds 915(66.35%) of respondents were from urban areas. Approximately half, (654, 47.43%) of the respondents had moderate stress. The majority, 1237(89.7%), of the respondents did not have a history of chronic illness (*Table 1*).

**Behavioral, academic, and relationship-related factors of participants.** A history of alcohol consumption was reported by more than one-third of the respondents (493, 35.75%). In terms of physical activity, only a minority of the respondents (224, 16.24%) reported being physically active. The majority of respondents (656, 47.57%) rated their academic ability as good. When it comes to perceived social support, approximately two-fifths of the respondents (564, 40.9%) reported low levels of perceived social support (**S1 Table**).

## Variables of the study

Outcome variables (endogenous constructs):

Depression, anxiety, and somatic symptoms

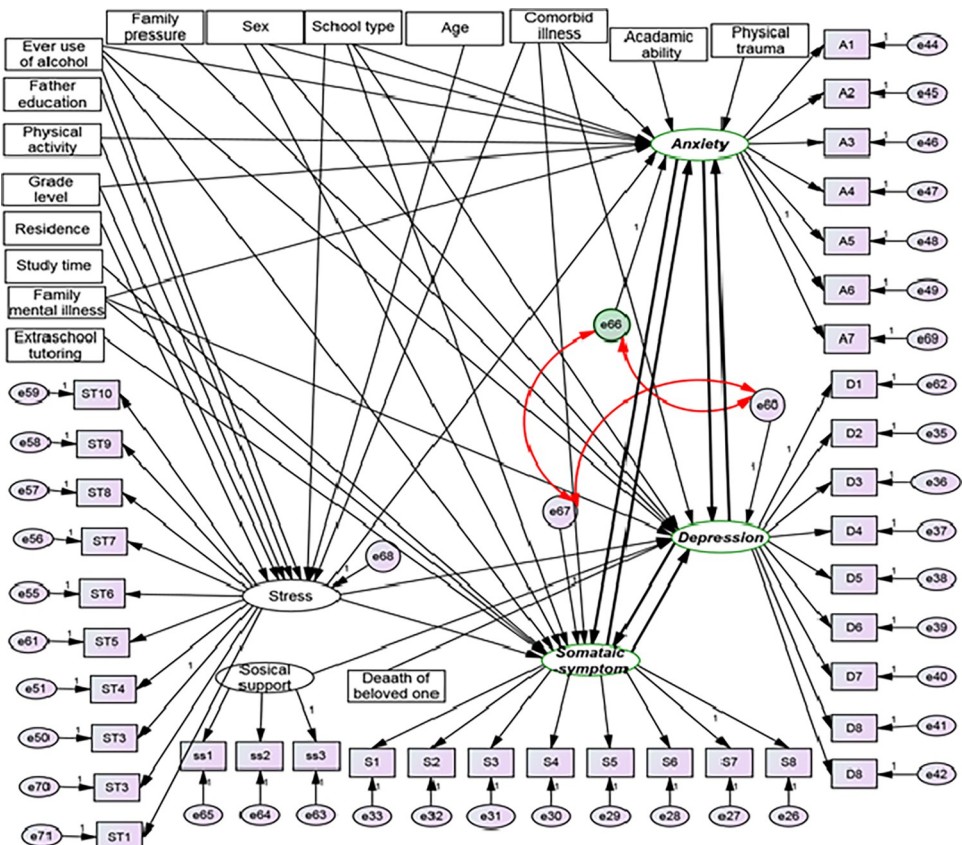

**Fig 1. Hypothetical structural equation modeling on the prevalence and determinants of depression, anxiety, and somatic symptoms among high school and preparatory school adolescents in Northwest Ethiopia, 2022 (both measurement and structural model): Circles indicate latent variables or error terms or disturbances, rectangles indicate observed variables, single arrows indicate factor loadings or regression coefficients, and double arrows indicate the covariance between latent variables.** A1-A7 = items of anxiety, D1-D9 = items for depression, S1-S8 = items for somatic symptoms, ST1-ST10 = items for stress, and e = error term.

**Independent variables:** summary of key variables used in the study (**S2 Table**).

## Data collection procedures and tools

Primary data in pen and pencil format were gathered using a structured questionnaire through a self-reported questionnaire for those who could see and by face-to-face interview for 3 individuals who could not see. Four trained data collectors with a first degree in public health were assigned for data collection. The data collection took place in a classroom setting, and it required approximately 15–25 minutes to complete the questionnaire.

The questionnaire was designed to gather data on socio-demographic characteristics, behavioral factors, academic-related factors, relationship-related factors, clinical factors, depression, anxiety, somatic symptoms, and stress domains (*Table 2*) (S1 Dataset). All questionnaire details were supplied in the supplementary information (*S3 Table*).

## Data quality assurance

To ensure the data's quality, several measures were taken. First, the questionnaire originally developed in English was translated into Amharic and then re-translated back into English by a different individual to ensure consistency. The face validity of the tool was reviewed by four

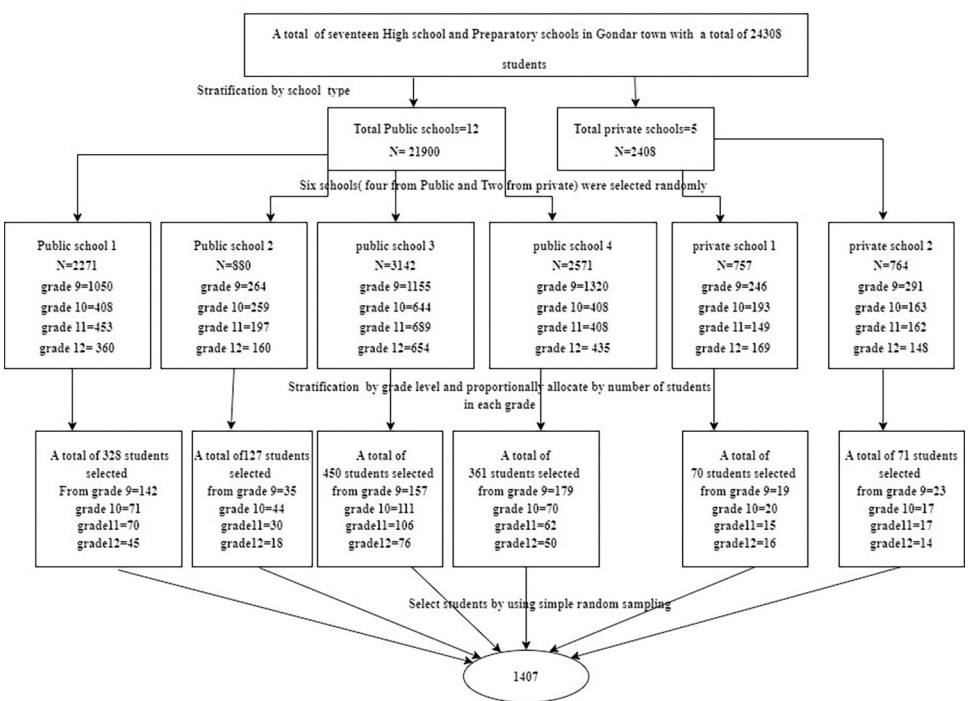

**Fig 2. Diagrammatical representation of the sampling procedure that was used to assess the prevalence and determinants of depression, anxiety, and somatic symptoms among high school and preparatory school adolescents in Northwest Ethiopia, 2022.** N indicates the total number of students.

public health experts and two psychiatrists. Reliability, construct validity, convergent validity, and omitted variable bias were all examined in the analysis process (**S1 Annex**).

## Data processing, model building, and analysis

Collected data were coded and entered into Epi-data software version 4.6, and exported to STATA version 16 and AMOS version 21 for further analysis. Non-recursive structural equation modeling was employed to assess the complex relationship between different latent and observed variables. Model assumptions such as: specified and identified model, multivariate normality, missing, outlier, sample size adequacy, independency, common method bias, and strong and valid instrumental variable were checked. Finally model fitness evaluation and model comparison were considered.

**Specified model**: In this study, we specified the model after extensive literature searching and reading as presented in (**Fig 1**).

**Identified Model**: In this study, non-recursive structural equation modeling with complete correlation of error terms was employed. There are three methods of identification in non-recursive structural equation modeling; which are: the presence of unique instrumental variables, the order condition, and the rank condition. Among these, only the rank condition is sufficient for identification, and the remaining two are necessary, but not sufficient [91]. In this study, all endogenous variables in the reciprocal loop had unique instrumental variables. Order condition: the number of excluded variables for that specific endogenous variable minus the total number of endogenous variables minus one should be greater than or equal to 0 [91]. In our model, to satisfy the order condition, each endogenous variable in the recursive loop should have at least 3–1 = 2 excluded variables. For depression, anxiety, and somatic

**Table 1. Socio-demographic and clinical characteristics of high and preparatory school adolescents in Northwest Ethiopia, 2022 (n = 1379).**

| Variable | Frequency | Percentage (%) |
|---|---|---|
| **Sex** | | |
| Male | 536 | 38.87 |
| Female | 843 | 61.13 |
| **Age** | | |
| 15 | 140 | 10.20 |
| 16 | 291 | 21.19 |
| 17 | 297 | 21.63 |
| 18 | 405 | 29.50 |
| 19 | 240 | 17.48 |
| **School type** | | |
| Public | 1238 | 89.78 |
| Private | 141 | 10.22 |
| **Grade Level** | | |
| Grade 9 | 544 | 39.45 |
| Grade 10 | 325 | 23.57 |
| Grade 11 | 296 | 21.46 |
| Grade 12 | 214 | 15.52 |
| **Residence** | | |
| Urban | 915 | 66.35 |
| Rural | 464 | 33.65 |
| **Physical trauma** | | |
| No | 1076 | 78.03 |
| Yes | 303 | 21.97 |
| **Medically confirmed chronic illness** | | |
| No | 1237 | 89.7 |
| Yes | 142 | 10.3 |
| **Stress** | | |
| Low | 653 | 47.35 |
| Moderate | 654 | 47.43 |
| High | 72 | 5.22 |
| **Family history of mental illness** | | |
| No | 1278 | 92.68 |
| Yes | 101 | 7.32 |

symptoms 9, 8, and 11variables were excluded respectively. Therefore, the model fulfills the requirement of the order condition. Rank condition: In this particular study, there are 3 endogenous variables in the non-recursive loop 17 exogenous variables, and one endogenous variable out of the feedback loop (*Fig 1*). Rank conditions were determined using the system matrix. To formulate the system matrix, a separate equation for each endogenous variable in the feedback loop is mandatory. Therefore, the equations for depression, anxiety, and somatic symptoms disorder are presented as follows:

*Depression*$(y1)$ = x2 alcohol use + x3 chronic illness + x4 physical activity + x5 family mental ilness+

x7 school ty + x11 death of beloved one + x12 social support + x16 family pressur

+ x18 stress + x20(y2) + x21(y3) somaticsymptom

**Table 2. Tools used for the determinants of depression anxiety, somatic symptom, their description and criteria for categorization.**

| Variables | Instruments | Description | Criteria for categorization | Reference |
|---|---|---|---|---|
| Depression | Patients Health Questionnaire 9 for adolescents (PHQ-9A) | It is a depression screening tool with 9 items; each item was measured by a Likert scale ranging from 0 "not at all" to 3 "nearly every day" and its total score ranged from 0 to 27. | Students were classified as having depression if their total score on the PHQ-9A was 10 or higher. | [84] |
| Anxiety | General Anxiety Scale-7 (GAS-7) | A brief measure of anxiety contains seven items, which are rated on four-point Likert scales that range from 0 "not at all" to 3 nearly every day". Its summed score ranges from 0–21. | Students were classified as anxious if their total score on the GAS-7 was 10 or higher. | [85] |
| Somatic symptom | Somatic symptom scale-8 (SSS-8) | It contains a total of 8 items, each item was measured by a Likert scale ranging from 0 "not at all" to 4"very much'; a total scoring range from minimal 0 to maximum 32. | Based on the SSS-8 score, students were classified as having low 4 to 7, medium 8 to 11, high 12 to 15, and very 16 to 32 very high somatic symptom levels. | [86] |
| Stress | Perceived Stress Scale 10 (PSS-10) | It contains 10 items. Each item scored with an ordinal scale of 0–4, where 0 represents "never" and 3 represents "very often". Individual scores range from 0 to 40. | Students who have a total perceived stress scale 10 scores of 0–13, 14–26, and 27–40 were categorized as having low, moderate, and high perceived stress levels respectively. | [87] |
| Social support | Oslo Social Support Scale three | It has 3 items; with a Likert scale that ranges from 1 to 4 for the first item (number of people you can count if you face a great personal problem) and 1 to 5 ranges for the rest items. It's summed score ranges from 3 to 14. | Students were categorized as having poor social support 3–8 score, moderate social support 9–11 score, and strong social support 12–14 score. | [88]. |
| Alcohol use | Alcohol, smoking, and substance involvement test(ASSIST) was used to assess current and ever use of alcohol | Do you have a history of alcohol (like beer, Tella, Katikala, Wine) intake in your lifetime? Do you drink alcohol within the past 3 months? | Respondents who answered yes were considered positive for ever-use of alcohol Students who answered yes were considered as positive for current alcohol user | [89] |
| Physical activity | Self-reported physical activity assessment for adolescents was used | In the past week, how many days did you exercise for at least 60 minutes until you felt sweaty or shortness of breath | Respondents who answered 2 days and above were categorized as physically active and those who answered one day and not were categorized as physically inactive. | [90] |

$$\textbf{\textit{Anxiety}}(y2) = x2 \text{ alcohol use} + x3 \text{ chronic illness} + x4 \text{ physical activity} + x5 \text{ family mental ilness} + x7 \text{ school type}$$
$$+ x13 \text{ physical trauma} + x17 \text{ acadamic ability} + x18 \text{ stress} + x19(y1) + x21(y3)$$

$$\textbf{Somaticsymptom}(\textbf{y3}) = x2 \text{ alcohol use} + x3 \text{ chronic illness} + x4 \text{ physical activity} + x5 \text{ family mental ilness} + x7$$
$$\text{school type} + x14 \text{ study time} + x15 \text{ extra school tutoring} + x18 \text{ stress} + x19(y1) + x20(y2)$$

Rank conditions in the system matrix below begin by constructing a system matrix based on the above equations. Endogenous variables are presented in the row and all variables are presented in the column. In each row, zero or one appears in the column that corresponds to that row. One indicates that the variable represented by the column has a direct effect on the endogenous variable in that row or on the endogenous variable itself, and 0 indicates the excluded variables for that specific endogenous variable in that row [91].

$$\begin{bmatrix} x & 2 & 3 & 4 & 5 & 6 & 7 & 8 & 9 & 10 & 11 & 12 & 13 & 14 & 15 & 16 & 17 & 18 & y1 & y2 & y3 \\ y1 & 1 & 1 & 1 & 1 & 0 & 1 & 0 & 0 & 0 & 1 & 1 & 0 & 0 & 0 & 1 & 0 & 1 & 1 & 1 & 1 \\ y2 & 1 & 1 & 1 & 1 & 0 & 1 & 0 & 0 & 0 & 0 & 0 & 1 & 0 & 0 & 0 & 1 & 1 & 1 & 1 & 1 \\ y3 & 1 & 1 & 1 & 1 & 0 & 1 & 0 & 0 & 0 & 0 & 0 & 0 & 1 & 1 & 0 & 0 & 1 & 1 & 1 & 1 \end{bmatrix}$$

The following steps were followed for the identification of rank condition

1. Start with the first row of the system matrix. Cross out all entries in that row and cross out any column in the system matrix with one in that same row.

2. Simplify the reduced matrix further by deleting any row with all zeros entries; delete any row that is an exact duplicate of another or can be reproduced by adding other rows (i.e., it is a linear combination of other rows). The number of remaining rows is the rank.

If the rank of the reduced matrix is larger than or equal to the sum of the endogenous variables in the recursive loop minus one, then the rank condition is satisfied [91].

The rank condition for this particular study became;

For depression (y1), after we cross the entire first row and the entire column that contains 1 in the row:

The reduced matrix becomes:

$$\begin{bmatrix} 1 & 0 \\ 0 & 1 \end{bmatrix} \text{Rank} = 2$$

For anxiety (y2), after we cross the entire second row and the entire column that contains 1 in that row:

The reduced matrix becomes:

$$\begin{bmatrix} 1 & 0 \\ 0 & 1 \end{bmatrix} \text{Rank} = 2$$

For Somatic symptom (y3), after we cross the entire third row and the entire column that contains one in that row:

The reduced matrix becomes:

$$\begin{bmatrix} 1 & 0 \\ 0 & 1 \end{bmatrix} \text{Rank} = 2$$

The hypothesized model was identified since all endogenous variables have a rank of two; which is equal to the number of endogenous variables in the feedback loop minus one.

**Multivariate normality, outliers, and missing:** Full information maximum likelihood estimation was used. Multivariate normality was assessed using Mardia's kurtosis and its critical ratio. The data were not normally distributed, since Mardia's kurtosis was above 7 and its critical ratio was above 5 [91]. Although item parceling and outlier deletion were attempted, there was no improvement. As a result, bootstrap maximum likelihood estimation with 3000 samples was used.

Mahalanobis distance p values less than 0.001 based on chi-square distribution were used to declare observation as a multivariate outlier [92]. There were 62 observations with Mahalanobis distance p-values less than 0.001. The data were examined to determine whether they were outliers or caused by data entry errors. However, this was not due to a data entry error, and as mentioned above, multivariate normality did not improve significantly with their exclusion. Therefore, those outlier observations were kept to maintain the study's power.

From the total observations, there were six observations with missing values. All missing values were from predictors and had no relationship with the outcome variable; thus, list-wise deletion was performed because they constituted less than 5% of the total sample and were considered missing completely at random, so 1373 observations were used for the final analysis [93].

**Sample size adequacy and strength of correlation:** The critical ratio for Bartlett's test of sphericity in this study was high and significant for all constructs. KMO was above 0.7 overall, as well as for specific constructs. Hence, the sample is sufficient, the population matrix was not identified, and factor analysis was evidenced [91] (*Table 3*).

**Table 3. KMO and Bartlett's test of sphericity for the measurement of depression, anxiety, and somatic symptoms among high and preparatory school adolescents in Northwest Ethiopia, 2022.**

| Factors | KMO | Bartlett's tests of sphericity | |
|---|---|---|---|
| | | Chi-square | P value |
| Anxiety | 0.88 | 3100 | 0.0001 |
| Depression | 0.87 | 2391 | 0.0001 |
| Somatic | 0.85 | 2155 | 0.0001 |
| Stress | 0.86 | 3354 | 0.0001 |
| Overall | 0.94 | 13938 | 0.0001 |

**Independency of observations**: The intra-class correlation coefficients (ICCs) were used to test the independence of observations. School type and grade level were employed as clustering variables. The ICC was less than 0.1 in our study (**S4 Table**), and a classical structural equation model was used.

**Common Method Bias:** Harman's one-factor test was used to check common method bias (CMB). There was no CMB in this study since the overall variance covered by one factor was 25.2%, which is below the suggested cut-off (50%) [94].

**Strong and valid instrumental variables**: The strength and validity of instrumental variables were checked by using the Cragg-Donald Wald F statistic and Sargan Hansen test, respectively. A Sargan Hansen test P-value greater than 0.05 was used to ponder valid instruments, and F-test values above 10 were used as a cut-off point to declare strong IVs [95]. In the present study, the strength and validity of the instrumental variables between anxiety and somatic symptoms, and between somatic symptoms and depression were invalid and weak. As a result, the non-recursive paths from somatic symptoms to anxiety and depression were excluded; since deleting a non-recursive path is one treatment for a weak and faulty instrument. Finally, an identified model with valid and strong instrumental variables in a non-recursive loop was preserved in (**S5 Table**).

**Model fitness and model comparison.** Confirmatory factor analysis (CFA) was conducted to evaluate the measurement model in the study. Model fit measures, including CMIN/DF, CFI, TLI, and RMSEA, were utilized to assess the overall goodness of fit of the model. However, in the hypothesized measurement model (**S1 Fig**), the goodness of fit fell below the acceptable threshold: CMIN/DF = 5.10, CFI = 0.81 and RMSEA = 0.05. To enhance the model fit, factorial item parceling was implemented [96] (**S6 Table**). Following this adjustment, the model fit improved, yielding CMIN/DF = 2.60, CFI = 0.99, TLI = 0.99, and RMSEA = 0.001 (**Fig 3**).

In the study, model comparisons were conducted for the structural and measurement components of the model using various criteria such as AIC, CMIN/DF, TLI, CFI, and RMSEA. The model that included only significant variables with significant disturbances after parceling was chosen as the well-fitted model. This selected model exhibited desirable characteristics, including a small AIC and CMIN/DF, a small RMSEA, and a large CFI and TLI (**Table 4**).

## Ethical consideration

Ethical approval was obtained from the Institutional Review Board of University of Gondar Institute of Public Health. Permission was obtained from school directors. Parental informed consent was waived as the study had no/minimal risk [97]. Following permission from the school directors, each student was given a detailed participant information sheet and they completed an assent form to indicate their willingness to participate. A total of twenty-eight non-volunteer students were identified and subsequently excluded from the study. Students

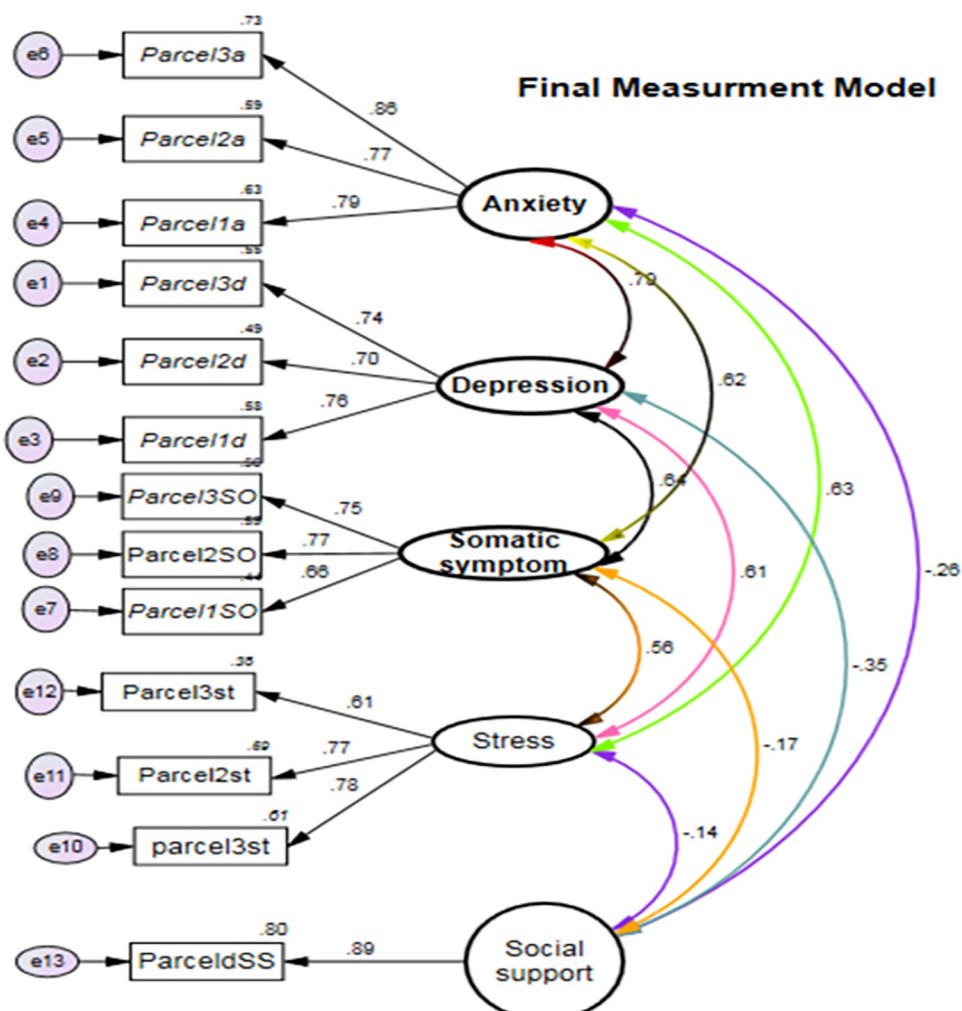

**Fig 3. Final measurement model for the constructs of anxiety, depression, somatic symptoms, stress, and social support. Key**: a = anxiety, d = depression, SO = somatic symptoms, st = stress, SS = social support, e = error term, and bidirectional arrow indicates covariance.

with moderate to severe anxiety, depression, or somatic symptoms were referred to the University of Gondar Comprehensive Specialized Hospital Psychiatry Clinic.

## Results

### Prevalence of depression

In this study, the overall prevalence of self-reported depression was 28.21(95% CI = 25.8, 31%). Regarding severity, 34.8% had mild depression, 18.5% had moderate depression, 7.72%

**Table 4. Model selection for determinants of depression, anxiety, and somatic symptoms among high and preparatory school adolescents in Northwest Ethiopia, 2022.**

| Models | AIC | CMIN/DF | CFI | TLI | RMSEA | Remark |
|---|---|---|---|---|---|---|
| A model with all predictors and covariance after the parcel | 2468 | 6.8 | 0.8 | 0.7 | 0.1 | |
| A model with only significant variables with significant disturbance after the parcel | 951 | 3.00 | 0.94 | 0.92 | 0.027 | **selected** |

had moderately severe depression and 2.04% o had severe depression. More than one-third,34.8% of participants had a feeling of hurting themselves or betting off dead, and more than two-thirds of the respondents (68.3%) had little interest or pleasure in doing things (**S7 Table**).

## Prevalence of anxiety

The overall prevalence of self-reported anxiety in this study was 25.05% (95%CI: 22.8, 27.5%). Regarding the level of anxiety, 34.08% of respondents had mild anxiety, 16.82% had moderate anxiety, and 8.27% had severe anxiety. Among the total respondents, 16.61% worried too much about different things nearly every day, and 11.54%were not able to stop worrying too much nearly every day. More than half (58.38%) of the respondents had a feeling of anxiety or nervousness (**S8 Table**).

## Prevalence of somatic symptoms

In this study, the overall prevalence of self-reported SSD was 25.24% (95% CI = 23, 27.6%). Regarding its level, among the total respondents, 27.27% had mild, 15.88% had moderate, 8.19% had high, and 6.38% had very high somatic symptoms. Regarding the response to specific indicators, approximately, two-thirds (65.42%) of the participants had headaches, more than one-half (53.68%) of the respondents had dizziness and 35.97% of the respondents had abdominal pain or gastrointestinal problem**s (S9 Table**).

## Relationship between depression, anxiety and somatic symptoms

There was a significant bidirectional relationship between anxiety and depression. Both anxiety and depression are related with high level of somatic symptoms. Anxiety had a direct positive effect [adjusted β = 0.74, 95% CI: 0.483, 1.081] on depression, and depression had a positive direct effect [adjusted β = 0.74, 95% CI: 0.508, 1.010] on anxiety. Depression had significant direct [adjusted β = 0.38, 95% CI = 0.167, 0.540] and indirect [adjusted β = 0.58, 95% CI: 0. 0.167, 3.629] positive effect on somatic symptoms resulting in a total positive effect of 0.96 [adjusted β = 0.96, 95% CI: 0.433, 1.456]. Anxiety had also a positive effect on somatic symptoms [adjusted β = 0.66, 95% CI: 0.270, 3.825].

## Factors related to depression, anxiety, and somatic symptoms

Self-rated academic ability, perceived social support, physical trauma, death of loved one, alcohol use, having medically confirmed chronic illness, sex, family pressure, school type, and stress were all significantly related to anxiety, depression, and somatic symptoms, either directly or indirectly. (*Fig 4*, *Tables 5–7*).

**Factors related to depression among high and preparatory school adolescents in Gondar town, 2022.** Perceived social support, having a history of the death of a loved one within the past six months, and anxiety had a significant direct effect on depression. However, having medically confirmed chronic illness, having a history of alcohol use, perceived family academic pressure, sex, stress, self-rated academic ability, and having a history of physical trauma had a statistically significant indirect relationship with depression. School type and anxiety were significantly related to depression both directly and indirectly.

Perceived social support [adjusted β = -0.13, 95% CI: -0.229,-0.029] had a positive direct effect on depression. Self-rated academic ability had a significant indirect effect [adjusted β = -0.05, 95% CI: -0.083,-0.031] on depression (*Table 5*).

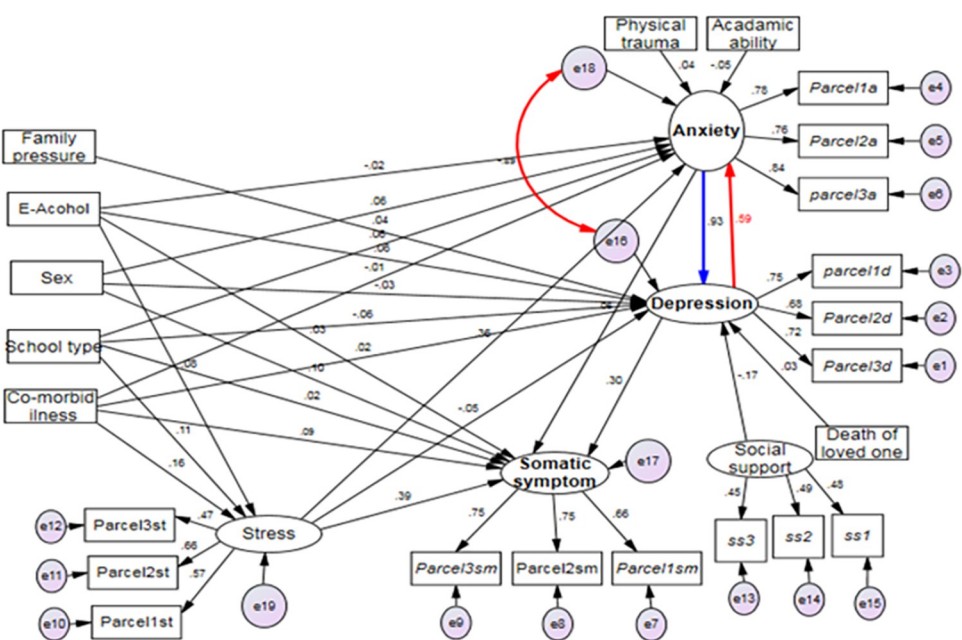

**Fig 4. Standardized structural equation modeling for factors related to anxiety, depression, and somatic symptoms among high and preparatory school adolescents in Gondar town, Northwest, Ethiopia, 2022.**

**Factors related to anxiety among high and preparatory school adolescents.** Self-rated academic ability and physical trauma had a significant direct effect on anxiety. However, alcohol use, and having a medically confirmed chronic illness had a significant indirect effect on anxiety. School type, sex, stress, and depression had statistically significant direct and indirect relationships with anxiety.

Having a history of physical trauma [adjusted $\beta$ = 0.06, 95% CI: 0.014, 0.125] had a positive direct effect on anxiety. Self-rated academic ability had a direct negative effect [adjusted $\beta$ = -0.03, 95% CI: -0.065, -0.006] on anxiety. Perceived social support had a significant negative

**Table 5. Direct, indirect, and total effects of socio-demographic, behavioral, relationship-related factors and clinical factors on depression among adolescents in Northwest Ethiopia, 2022: Unstandardized estimate.**

| Variables | Direct effect | Indirect effect | Total effect |
|---|---|---|---|
| DV: Depression | Estimate (95%CI) | Estimate(95% CI) | Estimate (95%CI) |
| Social support | -0.13[-0.229,-0.029] * | —— | -0.13[-0.229,-0.029] * |
| Death of a loved one | 0.03[0.004, 0.087] * | ——— | 0.03[0.004, 0.087] * |
| Self-rated academic ability | —— | -0.05[-0.083, -0.031]* | -0.05[-0.083, -0.031]* |
| Physical trauma | —— | 0.10[0.037, 0.166] * | 0.10[0.037, 0.166] * |
| Alcohol use | 0.06[-0.004, 0.112] | 0.08[0.029, 0.139] * | 0.14[0.073, 0.201]* |
| History of chronic illness | 0.04[-0.053, 0.124] | 0.17[0.085, 0.265] * | 0.21[0.114, 0.311]* |
| Sex | -0.03[-0.099, 0.025] | 0.09 [0.042,0.162] * | 0.06[0.003, 0.109]* |
| School type | -0.09[-0.194, -0.004]* | 0.20[0.113, 0.320] * | 0.11[0.016, 0.207]* |
| Family pressure | 0.02[-0.004, 0.040] | 0.02[0.002, 0.043] * | 0.04[0.016, 0.060]* |
| Stress | -0.06[-0,454, 0.247] | 0.82[0.530, 1.250] * | 0.76[0.642, 0.900] * |
| Anxiety | 0.74[0.483, 1.081*] | | 0.74[0.483, 1.081*] |

*Indicates significant variables, DV = dependent variable, SE = standard error, and CI = confidence interval.

**Table 6. Direct, indirect, and total effect of socio-demographic, behavioral, relationship-related, and clinical factors on anxiety among adolescents in Northwest Ethiopia, 2022: Unstandardized estimate.**

| Variables | Direct effect | Indirect effect | Total effect |
|---|---|---|---|
| DV: Anxiety | Estimate (95% CI) | Estimate (95%CI) | Estimate (95% CI) |
| Self-rated academic ability | -0.03[-0.065, 0.006] * | —— | -0.03[-0.065, 0.006] * |
| Physical trauma | 0.06[0.014, 0.125]* | —— | 0.06[0.014, 0.125] * |
| Social support | —— | -0.22[-0.304,-0.139] * | -0.22[-0.304,-0.139] * |
| Death of a loved one | —— | 0.06[0.010, 0.122] * | 0.06[0.010, 0.122] * |
| Alcohol use | -0.03[0.088, 0.035] | 0.14[0.071, 0.213] * | 0.11[0.041, 0.188] * |
| History of chronic illness | -0.02[-0.124, 0.078] | 0.27[0.165, 0.384] * | 0.25[0.132, 0.371] * |
| Sex | 0.08[0.019, 0.135] * | 0.04[0.004, 0.092] * | 0.12[0.059, 0.179] * |
| School type | 0.12[0.017, 0.226] * | 0.16[0.056, 0.269] * | 0.28[0.159, 0.415] * |
| Family academic pressure | —— | 0.03[0.012, 0.050] * | 0.03[0.001, 0.50] * |
| Stress | 0.54[0.293, 0.745] * | 0.57[0.379, 0.814] * | 1.10[0.955, 1.264] * |
| Depression | 0.74[0.508, 1.010] * | —— | 0.74[0.508, 1.010] * |

*Indicates significant variables, DV = dependent variable, SE = standard error, and CI = confidence interval.

indirect effect [adjusted β = -0.22, 95% CI: -.304, -0.139] on anxiety. Stress had both direct [adjusted β = 0.54, 95% CI: 0.293, 0.745] and indirect [adjusted β = 0.57, 95% CI: 0.379, 0.814] positive effect on anxiety, bringing a total positive effect of 1.10[adjusted β = 1.10, 95% CI: 0.955, 1.264] (*Table 6*).

**Factors related to somatic symptoms among high and preparatory school adolescents.**
Self-rated academic ability, perceived social support, physical trauma, and the death of a loved one in the past 6 months had a significant indirect effect on somatic symptoms. However, having a medically confirmed chronic illness, sex, stress, and depression were all directly and indirectly related to somatic symptoms.

Social support [adjusted β = -0.11, 95% CI: -0.176, -0.067] and self-rated academic ability [adjusted β = -0.02, 95%CI: -0.039, -010] had a significant negative indirect effect on somatic symptoms. Having medically confirmed chronic illness had significant direct [adjusted β =

**Table 7. Direct, indirect, and total effects of socio -demographic, personal, relationship-related factors, and stress on anxiety, depression, and somatic symptom disorders among adolescents in Northwest Ethiopia, 2022: Unstandardized estimate.**

| Variables | Direct effect | Indirect effect | Total effect |
|---|---|---|---|
| DV: Somatic symptoms | Estimate (95%) | Estimate (95%) | Estimate (95%) |
| Self-rated academic ability | —— | -0.02[-0.039, -010]* | -0.02[-0.039, -010]* |
| Physical trauma | —— | 0.04[0.012, 0.083] * | 0.04[0.012, 0.083]* |
| Social support | —— | -0.11[-0.176, -0.067]* | -0.11[-0.176, -0.067]* |
| Death of a loved one within the past 6 months | —— | 0.03[0.005, 0.067] * | 0.03[0.005, 0.067] * |
| Alcohol use | 0.03[-0.022, 0.105] | 0.09[0.044, 0.144] * | 0.12[0.062, 0.211]* |
| History of medically confirmed chronic illness | 0.17[0.045, 0.290]* | 0.20[0.122, 0.289]* | 0.37[0.239, 0.505] * |
| Sex | 0.11 [0.050, 0.166]* | 0.03[0.002, 0.053]* | 0.14[0.074, 0.198] * |
| School type | 0.03[-0.074, 0.144] | 0.13[0.053, 0.216]* | 0.16[0.046, 0.292]* |
| Stress | 0.53[0.326, 0.768]* | 0.33[0.186, 0.472] * | 0.86[0.700, 1.025]* |
| Depression | 0.38[0.167, 0.540]* | 0.58[0.167, 3.629] * | 0.96[0.433, 3.774] * |
| Anxiety | 0.06[-0.101, 0.212] | 0.60[0.243, 3.825]* | 0.66[0.270, 3.825]* |

*Indicates significant variables, DV = dependent variable, SE = standard error, and CI = confidence interval.

0.17, 95% CI = 0.045, 0.290] and indirect [adjusted β = 0.20, 95% CI: 0.122, 0.289] effects on somatic symptoms, bringing a total positive effect of 0.37 [adjusted β = 0.37, 95% CI: 0.239, 0.505]. Stress had both direct [adjusted β = 0.53, 95% CI: 0.326, 0.768] and indirect [adjusted β = 0.33, 95% CI: 0.186, 0.472] positive effect on somatic symptoms resulting in a total positive effect of 0.86 [adjusted β = 086, 95% CI: 0.700, 1.025] (*Table 7*).

## Discussion

In this study, the prevalence and determinants of depression, anxiety, and somatic symptoms and their relationships were examined using non-recursive structural equation modeling. The prevalence of depression, anxiety and somatic symptoms were moderate. Sex, alcohol use, stress, self-rated academic ability, perceived social support, school type, and physical trauma were significant determinants of anxiety, depression, and somatic symptoms. The bidirectional relationship between anxiety and depression was significant. Depression and anxiety were significant determinants of somatic symptoms.

### Prevalence of depression, anxiety, and somatic symptoms

In the current study, the overall prevalence of self-reported depression was 28.21% (95% CI = 25.8%, 31%). This is in line with the study conducted in Jimma 28% [36], but it was higher than the meta-analysis conducted in China (24.3%) [33], and lower than the studies conducted in Aksum (38.2%) [37], Nepal (44.2%) [98], Bangladesh (36%) [99], and India(57.7%) [100]. In this study, the prevalence of anxiety was 25.05% (95%CI: 22.8, 27.5%). This is consistent with the study conducted among children and youth in Ethiopia 0.5 to 23% [20]. However, it is lower than the study conducted in Kenya at 37.99% [19], Saudi Arabia at 63.5% to 66% [101,102], Jordan at 42.1% [17], and Chandigarh at 80.85% [18]. In contrast, it was higher than the worldwide estimates of anxiety among children and adolescents (6.5 to 10%). In our study, the prevalence of self-reported somatic symptom disorders was 25.24% (95% CI = 23, 27.6%). This is consistent with the study conducted among children and adolescents in Tarragona which reported 5 to 30% [62]. However, it was higher than the prevalence in the general population (5–7%) [103] and lower than that in a study conducted in Qatar (47.8% in females and 52.2% in males) [104]. The variation in the prevalence of depression, anxiety, and somatic symptoms across different settings can potentially be attributed to differences in screening tools and the choice of cutoff points used for assessment [18,37,105]. Furthermore, since mental health is the combination of physical, social, cultural, and religious environments this could be explained by socio-cultural differences in different settings. Poverty, violence, and other stressful social conditions are not unique to any one region of the world, nor are the symptoms and manifestations that result from them. However, factors that are frequently associated with race or ethnicity, such as socioeconomic position or country of origin, can enhance the likelihood of being exposed to these sorts of stressors, and thus stressors may lead to depression, anxiety, and somatic symptoms [106]. Furthermore, the ongoing internal conflict in northern Ethiopia may contribute to an increased prevalence of depression, anxiety, and somatic symptoms. The exposure to traumatic events, displacement, and the disruption of social support systems can have profound psychological effects on individuals living in conflict-affected areas. The constant fear, uncertainty, and instability associated with conflict can lead to heightened levels of stress and psychological distress, resulting in a higher incidence of mental health issues [81].

### Relationship between anxiety, depression and somatic symptoms

After controlling for other factors, we observed that depression had a direct positive effect on anxiety [adjusted β = 0.74, 95% CI: 0.508, 1.010]. This finding aligns with a study conducted in

the United Kingdom [30]. It is possible that individuals with depression spend a significant amount of time worrying about their symptoms, leading to increased anxiety. Additionally, the co-occurrence of depression and anxiety could be influenced by a shared set of genes [107]. Furthermore, we observed that high level of anxiety was also significantly related to higher level of depression [adjusted β = 0.74, 95% CI: 0.483, 1.081]. This finding is compatible with other findings [53,54]. One possible explanation for this association is that individuals with anxiety may attempt to control their worries. However, if they are unable to effectively manage their anxiety, it can negatively impact their emotions, leading to feelings of sadness, hopelessness, and depression. Additionally, this relationship could be influenced by the presence of shared genetic factors and environmental influences [107].

As the level of depression increases the level of somatic symptoms will also increase [adjusted β = 0.96, 95% CI = 0.167, 3.629]. In the same way, as the level of anxiety increases the level of somatic symptoms also increases [adjusted β = 0.66, 95% CI = 0.270, 3.825]. This result is supported by other studies [66,74–77]. A possible explanation for this consistency could be that in depressed individuals' neurotransmitters such as nor-epinephrine and serotonin are low; a decrease in neurotransmitters may lead individuals to focus on physical symptoms. Moreover, it could be explained by the co-occurrence of somatic symptoms, anxiety, and depression. The sharing of common sets of gene and environmental factors may also be a possible justification.

## Factors related to depression, anxiety, and somatic symptoms

Female sex had a statistically significant positive effect on depression [adjusted β = 0.06, 95% CI: 0.003, 0.109], anxiety [adjusted β = 0.12, 95% CI: 0.059, 0.179], and somatic symptoms [adjusted β = 0.13, 95% CI: 0.074, 0.198]; this implies that being female increases the levels of depression, anxiety, and somatic symptoms compared to being male, controlling for other factors. This finding is consistent with other studies [36,38–40]. The observed differences between men and women can be attributed to various factors, including variances in brain chemistry and hormonal fluctuations. Women may experience changes in neurotransmitters due to menstrual fluctuations in progesterone and estrogen, which can contribute to symptoms of depression, anxiety, and somatic complaints. Additionally, the presence of androgen receptors in males may provide some protective effects against certain mental health disorders [108]. Another influencing factor could be the differences in reporting behavior between genders. Females may be more inclined to report and seek help for mental illnesses compared to males. Cultural factors also play a role in how men and women approach and disclose their experiences with depression and other mental health conditions. These cultural influences can shape the willingness of individuals to openly discuss and seek support for their mental health challenges [65,66,109–111]. Gender role might also be the possible reason: Girls are often socialized to prioritize traits such as being nurturing, accommodating, and conforming to societal expectations. These expectations can create pressure to meet unrealistic standards, leading to increased stress and vulnerability to mental illness.

Self-rated academic ability, being in private school and perceived social support had significant association with depression, anxiety and somatic symptoms. As the self-rated academic ability of adolescents increases, the levels of depression [adjusted β = -0.05, 95% CI: -0.083,-0.031], anxiety [adjusted β = -0.03, 95% CI: -0.065, -0.006], and somatic symptoms [adjusted β = -0.02, 95% CI: -0.039, -010] decrease. This result is compatible with the study conducted in Iran [112], and Slovakia [113] and with other studies [17,25,114]. The possible explanation for this might be that students with low academic ability are terrified of negative responses from teachers, parents, and friends, which may lead to fear about their future career, which will lead

them to become anxious, and anxiety will lead to depression and somatic symptoms. Learning in a private school had a statistically significant positive effect on depression [adjusted β = 0.10, 95% CI: 0.016, 0.207], anxiety [adjusted β = 0.28, 95% CI: 0.159, 0.415 and somatic symptoms [adjusted β = 0.16, 95% CI: 0.046, 0.292]. This means that participants in private schools had higher levels of depression, anxiety, and somatic symptoms than their counterparts. In contrast to our finding, the study conducted in India [41] showed that public school students had a higher level of depression than private school students; this disparity may be explained by differences in sample size and method of analysis. Our study had a larger sample size than the India study, and the India study did not address the confounding effect. Higher levels of depression, anxiety and somatic symptoms in private schools may be due to parents' expectations of high academic achievement in private schools; which leads students to be stressed when they try to meet their parents' expectations, and stress leads to depression, anxiety, and somatic symptoms. Perceived social support had a significant negative effect on depression [adjusted β = -0.13, 95% CI: -0.229,-0.029], anxiety [adjusted β = -0.22, 95% CI: -.304, -0.139], and somatic symptoms [adjusted β = - 0.11, 95% CI: -0.176, -0.067]. This means that as social support increased, the level of depression, anxiety and somatic symptoms decreased or students with low social support had higher levels of depression, anxiety, and somatic symptoms than those who had high social support. This finding is compatible with different reports [43,44,51,55]. The probable reason for this might be that low social support may increase feelings of loneliness, worthlessness, and hopelessness, leading to a loss of interest in activities and depression, depression might leads to anxiety, and somatic symptom. Moreover, students with low social support are more likely to have low self-esteem and negative cognition than those with higher social support, which may lead to depression [115].

Stress had a significant positive effect on depression [adjusted β = 0.76, 95% CI: 0.642, 0.900], anxiety [adjusted β = 1.10, 95% CI: 0.955, 1.264], and somatic symptoms [adjusted β = 086, 95% C: 0.700, 1.025]. This indicates that participants with high levels of stress had higher levels of depression, anxiety, and somatic symptoms, holding other predictors constant. Our finding is supported by different studies [48,49,71]. The possible justification for this could be that high-level stress impairs brain function, which may lead to fluctuations in neurotransmitters such as dopamine, which leads to depression. In addition, immune dysregulation during stressful life events leads to anxiety, and anxiety leads to depression and depression may lead to somatic symptoms.

Alcohol use had a positive effect on depression [β = 0.14, 95% CI: 0.073, 0.201], anxiety [adjusted β = 0.11, 95% CI: 0.041, 0.188], and somatic symptoms [adjusted β = 0.12, 95% CI: 0.062, 0.211]. This implied that adolescents who had a history of alcohol drinking had higher levels of depression, anxiety, and somatic symptoms than those who did not drink. This finding is congruent with the study conducted in the USA [45] and in China [46,116]. This could be because alcohol affects brain chemicals such as serotonin and dopamine, which are responsible for happiness, and the decrease in these chemicals leads to feelings of depression [117]. Furthermore, alcohol consumption may contribute to sadness, anxiety, and somatic symptoms in adolescents by reducing age-appropriate activities such as physical activity and social relationships [118]. Moreover, it could be due to the direct effect of ethanol on nerve cells [119] which leads to heightened awareness of normal body sensations. It is also worth noting that alcohol use can be a potential symptom of these disorders.

Our findings showed that holding other predictors constant, having a history of physical trauma had a direct positive effect [adjusted β = 0.06, 95% CI: 0.014, 0.125] on anxiety. This indicates that students with physical trauma had higher levels of anxiety than those who did not. Our finding was supported by the study conducted in the Netherlands [120]. This consistency could be justified by the fact that in reaction to trauma, neurotransmitters such as

dopamine, serotonin, and gamma amino butyric acid may be impacted by cortisol and depletion of these neurotransmitters may lead to anxiety. In addition, individuals with physical trauma may internalize the event and may have subsequent negative thoughts about themselves, which affect their emotional experience and might lead to anxiety [121].

## Strengths and limitations

The study examined a bidirectional relationship between anxiety and depression, considering both direct and indirect effects of various independent variables on different dependent variables. The sample included adolescents from both private and public schools, focusing on an underrepresented segment of the population. However, it is important to acknowledge that this study is not without limitations. First, only school adolescents in Northwest Ethiopia were included in the study which may affect generalization to all adolescents in Ethiopia. Second, self-reported screening tools were used, which may overestimate the effect size. Third, we only asked about alcohol use in the last 3 months, not how much, how frequently, or for how long, which could influence the outcome. Last, the temporal relationship between predictors and outcomes was not evaluated due to the cross-sectional nature of the data.

## Conclusion

In this study, a moderate prevalence of depression, anxiety, and somatic symptoms was found. Self-rated academic ability, physical trauma, school type, sex, stress, alcohol use, and perceived social support were significant determinants of depression, anxiety, and somatic symptoms. The bidirectional relationship between anxiety and depression was significant. Depression and anxiety were significant determinants of somatic symptoms. Based on the study findings, the authors suggest the following points: Policymakers should prioritize bolstering the prevention and management of mental health disorders among adolescents. One effective approach is to introduce and establish school-based mental health screening services. It is crucial to recognize and address the bidirectional relationship between anxiety and depression, while also considering somatic symptoms as potential indicators in individuals with these conditions. Additionally, conducting studies that encompass out-of-school adolescents is highly recommended. Furthermore, to gain a deeper understanding of the temporal relationships involved, follow-up studies should be conducted.

## Supporting information

**S1 Fig. Hypothetical measurement modelfor the determinants of adolescents' depression, anxiety, and somatic symptoms in Northwest Ethiopia, 2022.**
(DOCX)

**S1 Table. Behavioral, academics, and relationship related factorsof adolescents' Northwest Ethiopia, 2022.**
(DOCX)

**S2 Table. Summary of key variables Determinants of adolescents' depression, anxiety, and somatic symptoms in Northwest Ethiopia, 2022.**
(DOCX)

**S3 Table. Tools for the determinants of adolescents' depression, anxiety, and somatic symptoms in Northwest Ethiopia, 2022.**
(DOCX)

**S4 Table. Intraclass correlation coefficients for depression, anxiet, and somatic symptoms among adolescents in Northwest Ethiopia, 2022.**
(DOCX)

**S5 Table. Instrumental variable validity and strength for the determinants of adolescents' depression, anxiety, and somatic symptoms in Northwest Ethiopia, 2022.**
(DOCX)

**S6 Table. Factorial average item parceling for the determinants of adolescents' depression, anxiety, and somatic symptoms in Northwest Ethiopia, 2022.**
(DOCX)

**S7 Table. Depression among adolescents in Northwest Ethiopia, 2022.**
(DOCX)

**S8 Table. Anxiety among adolescents in Northwest Ethiopia, 2022.**
(DOCX)

**S9 Table. Somatic symptoms among adolescents in Northwest Ethiopia, 2022.**
(DOCX)

**S1 Annex. Result of the pilot study for the determinants of adolescents' depression, anxiety, and somatic symptoms in Northwest Ethiopia, 2022.**
(DOCX)

**S1 Dataset. Dataset for the determinants of depression, anxiety, and somatics symptoms among adolescents in Gondar town 2022.**
(XLS)

## Acknowledgments

The authors are grateful to the University of Gondar, data collectors, supervisors, and study participants.

## Author Contributions

**Conceptualization:** Zenebe Abebe Gebreegziabher.

**Data curation:** Zenebe Abebe Gebreegziabher, Rediet Eristu, Ayenew Molla.

**Formal analysis:** Zenebe Abebe Gebreegziabher.

**Funding acquisition:** Zenebe Abebe Gebreegziabher, Rediet Eristu, Ayenew Molla.

**Investigation:** Zenebe Abebe Gebreegziabher, Rediet Eristu, Ayenew Molla.

**Methodology:** Zenebe Abebe Gebreegziabher, Rediet Eristu, Ayenew Molla.

**Project administration:** Zenebe Abebe Gebreegziabher, Rediet Eristu, Ayenew Molla.

**Resources:** Zenebe Abebe Gebreegziabher, Rediet Eristu, Ayenew Molla.

**Software:** Zenebe Abebe Gebreegziabher, Rediet Eristu, Ayenew Molla.

**Supervision:** Zenebe Abebe Gebreegziabher, Rediet Eristu, Ayenew Molla.

**Validation:** Zenebe Abebe Gebreegziabher, Rediet Eristu, Ayenew Molla.

**Visualization:** Zenebe Abebe Gebreegziabher.

**Writing – original draft:** Zenebe Abebe Gebreegziabher.

**Writing – review & editing:** Zenebe Abebe Gebreegziabher, Rediet Eristu, Ayenew Molla.

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
