## [Decision Letter · Decision Letter 0]

19 Apr 2023

PONE-D-23-02244Depression, Anxiety, Somatic symptom and their determinants among High School and Preparatory School Adolescents  in Gondar Town, Northwest Ethiopia, 2022.Non-recursive Structural Equation Modeling.PLOS ONE

Dear Dr. Gebreegziabher,

Thank you for submitting your manuscript to PLOS ONE. After careful consideration, we feel that it has merit but does not fully meet PLOS ONE’s publication criteria as it currently stands. Therefore, we invite you to submit a revised version of the manuscript that addresses the points raised during the review process.

We look forward to receiving your revised manuscript.

Kind regards,

Ching-Fang Sun, MD

Academic Editor

PLOS ONE

Additional Editor Comments:

The study is well-designed. However, the expression style has to be modified for better clarity.

Please adjust the reviewers' comments. (See attachment from reviewer 2)

Reviewers' comments:

Reviewer's Responses to Questions

**Comments to the Author**

1. Is the manuscript technically sound, and do the data support the conclusions?

Reviewer #1: Yes

Reviewer #2: Yes

2. Has the statistical analysis been performed appropriately and rigorously? 

Reviewer #1: Yes

Reviewer #2: Yes

3. Have the authors made all data underlying the findings in their manuscript fully available?

Reviewer #1: Yes

Reviewer #2: Yes

4. Is the manuscript presented in an intelligible fashion and written in standard English?

Reviewer #1: Yes

Reviewer #2: Yes

5. Review Comments to the Author

Reviewer #1: General Feedback!

The write up, grammar and integrity is good

Eligibility criteria

Was there no exclusion criteria used?

Operational Definition

Do you consider the person with PHQ-9A score of less than 9 have no depression? A person with a PHQ-9A score of 5-9 has mild depression. Do you think missing these segments of the population has effect on your result?

Strength and Limitation of the study

‘Assed’: Good to write in full assessed on line 955

Reviewer #2: To the best of my knowledge the study was well executed, the data sound and the analyses appropriate (though I leave the final judgement on analysis to reviewers more expert than me in SEM). The manuscript is on an important topic, it is reasonably well-written and the study findings ought to be brought to light. However, considerable re-organization and making the paper far more concise will be required to meet the standards for publication, in any journal.

6. PLOS authors have the option to publish the peer review history of their article (what does this mean?). If published, this will include your full peer review and any attached files.

Reviewer #1: **Yes: **Dr Anteneh Messele Birhanu

Reviewer #2: No

---

## [Author Response · Author response to Decision Letter 0]

22 Jun 2023

Thank you for submitting your manuscript to PLOS ONE. After careful consideration, we feel that it has merit but does not fully meet PLOS ONE’s publication criteria as it currently stands. Therefore, we invite you to submit a revised version of the manuscript that addresses the points raised during the review process.

---

## [Editor Report · Decision Letter 1]

17 Jul 2023

PONE-D-23-02244R1Determinants of  adolescence depression, anxiety and somatic symptom in northwest EthiopiaPLOS ONE

Dear Dr. Zenebe Abebe Gebreegziabher,

Thank you for submitting your manuscript to PLOS ONE. After careful consideration, we feel that it has merit but does not fully meet PLOS ONE’s publication criteria as it currently stands. Therefore, we invite you to submit a revised version of the manuscript that addresses the points raised during the review process.

 Please see detailed comments as below and attached.

Please submit your revised manuscript by Aug 31 2023 11:59PM**. **If you will need more time than this to complete your revisions, please reply to this message or contact the journal office at plosone@plos.org. Please include the following items when submitting your revised manuscript:

We look forward to receiving your revised manuscript.

Kind regards,

Ching-Fang Sun, MD

Academic Editor

PLOS ONE

Journal Requirements:

**Additional Editor Comments:**

**Title:**

Please change the title into “…Ethiopia**: A** Non-recursive structural…”

**Abstract:**

Please see the recommended change as attached.

**Methodology and Discussion:**

I agree with the paper organization issue pointed out by reviewer 2. The current version is still satisfactory. Please see detailed comments as attached.

**Discussion:**

Please condense the 3 paragraphs in “Magnitude of depression, anxiety and somatic symptoms” into a single paragraph. It is unnecessary for the author to repeat the same logic three times despite different outcomes (report finding, compare findings with multiple previous studies, explained the difference might be a result of screening tools). It is also unnecessary for the author to elaborate on details of previous studies. The reviewer 2 pointed out the result is too long. So is discussion. Please see the rest of comments as attached.

**General:**

Specific terms need to be better defined:The author used the term “magnitude” and “prevalence” alternatively to describe the outcomes including depression, anxiety, and somatic symptoms. Please clarify the definition of these two terms and use the correct one. The author may find it beneficial to consult a statistician for better description.Participants:Please define the participant’s age in both the text and tables. (Grades are not satisfactory in this case) If the author includes any participants aged under 12, the participant description should be revised into “children and adolescent” through the whole manuscript.Logic of clinical application:It is suggested that the author invite a clinical practitioner working with the investigated population to go through the study. The author highlights the correlation between depression-anxiety-somatic symptoms as the main outcome; alcohol use, sex and more are correlated with unfavorable mental health situation. However, these findings did not provide novelty to the field. The value of this study is the unique study population. The author could have better elaborate how the results could be explained by environmental factors specific in Northwest Ethiopia and how that could affect policy making/clinical practice.Writing quality:Please correct all the typos, lower case/upper case, repetitive content, inconsistent font, content generated during the reversing process which supposed to be deleted from the formal manuscript. These are the red flags for the editor that suggested lacking effort in proofreading.Language:The syntax is confusing. Please consider having a senior author or peer to help with better expression.

Reviewers' comments:

NA (Please address the editor's comment. Reviewers will be re-invited once you re-submit your manuscript.)

---

## [Author Response · Author response to Decision Letter 1]

16 Aug 2023

1. Response to chief academic editor 

Manuscript title: Depression, Anxiety, somatic symptom and their determinants among high school students in North West Ethiopia: Non-recursive structural equation modeling 

Manuscript ID: PONE-D-23-02244

Authors’ response: Dear Editors and Reviewers, Thank you for your constructive feedback regarding our manuscript. The authors are appreciative of the editors' and reviewers' helpful criticism, insight, time, and positive assessments of our work. Based on the chief editor's and reviewers' comments, we made numerous new and clarifying statements throughout the entire document. The revised text was marked and attached as a separate file. These changes have improved the manuscript considerably and we hope that it can be published without delay. All authors have read the revised version of the manuscript and do not have any conflict of interest. We appreciate the editors and reviewers for their time and critical comments to improve our manuscript and we addressed the comments point by point as follows. We hope the revised manuscript satisfies the chief editor as well respected reviewers. If there are any unresolved questions, please do not hesitate to contact us again. 

Editor: Journal requirements:

Author response: We reviewed and corrected the formatting in accordance with the PLOS ONE criteria.

Editor: 2. In your Data Availability statement, you have not specified where the minimal data set underlying the results described in your manuscript can be found. PLOS defines a study's minimal data set as the underlying data used to reach the conclusions drawn in the manuscript and any additional data required to replicate the reported study findings in their entirety. All PLOS journals require that the minimal data set be made fully available. For more information about our data policy, please see http://journals.plos.org/plosone/s/data-availability.

Author response: Dear PLOSONE journal editors, thank you for your suggestion to include the data availability statement in the manuscript. We accepted your comment and included it in the supplementary file on the revised submission.

Editor: If applicable, we recommend that you deposit your laboratory protocols in protocols.io to enhance the reproducibility of your results. Protocols.io assigns your protocol its own identifier (DOI) so that it can be cited independently in the future. For instructions see: https://journals.plos.org/plosone/s/submission-guidelines#loc-laboratory-protocols. Additionally, PLOS ONE offers an option for publishing peer-reviewed Lab Protocol articles, which describe protocols hosted on protocols.io. Read more information on sharing protocols at https://plos.org/protocols?utm_medium=editorial-email&utm_source=authorletters&utm_campaign=protocols.

Authors’ response: not applicable.

Editor: 3. Please review your reference list to ensure that it is complete and correct. If you have cited papers that have been retracted, please include the rationale for doing so in the manuscript text, or remove these references and replace them with relevant current references. Any changes to the reference list should be mentioned in the rebuttal letter that accompanies your revised manuscript. If you need to cite a retracted article, indicate the article’s retracted status in the References list and also include a citation and full reference for the retraction notice.

Authors’ response: We appreciate the editors' for their use full insight to check the presence of retracted papers in our reference list, we revised our references and there is no reference list from retracted paper in our manuscript.

Additional Editor Comments:

The study is well-designed. However, the expression style has to be modified for better clarity.

Please adjust the reviewers' comments. (See attachment from reviewer 2)

Authors’ response: Thank you for your advice on how to make our work better. We have incorporated these into the revised manuscript along with the second reviewers' ideas, and the language used throughout the manuscript, notably in the result and discussion sections has been amended accordingly.

Title:

Editor: Please change the title into “…Ethiopia: A Non-recursive structural…”

Authors’ response: Accepted and corrected (line number 2-3).

Abstract:

Editor: Please see the recommended change as attached.

Authors’ response: Thank you for the suggestion, we accepted it and modified accordingly (line number 12, 25, 33).

Methodology and Discussion:

Editor: I agree with the paper organization issue pointed out by reviewer 2. The current version is still satisfactory. Please see detailed comments as attached.

Authors’ response: We appreciate your feedback on how to make our paper better, and we have addressed all of your concerns in the amended version (example: line number 45-47, 151,217 and like).

Discussion:

Editor: Please condense the 3 paragraphs in “Magnitude of depression, anxiety and somatic symptoms” into a single paragraph. 

Authors’ response: Accepted and modified (line number 441-452).

Editor: It is unnecessary for the author to repeat the same logic three times despite different outcomes (report finding, compare findings with multiple previous studies, explained the difference might be a result of screening tools). It is also unnecessary for the author to elaborate on details of previous studies. The reviewer 2 pointed out the result is too long. So is discussion. Please see the rest of comments as attached

Authors’ response: Accepted and modified (line number 454-462).

General:

Editor: Specific terms need to be better defined:

The author used the term “magnitude” and “prevalence” alternatively to describe the outcomes including depression, anxiety, and somatic symptoms. Please clarify the definition of these two terms and use the correct one. The author may find it beneficial to consult a statistician for better description.

Authors’ response: We appreciate the wise advice you provided. We consulted statistician regarding the terms magnitude and prevalence; in our instance, prevalence was the more acceptable term, thus we replaced magnitude with prevalence throughout the entire document (Example: line number 25, 33, 331, 337, and 343).

Editor: Participants:

Please define the participant’s age in both the text and tables. (Grades are not satisfactory in this case) If the author includes any participants aged under 12, the participant description should be revised into “children and adolescent” through the whole manuscript.

Authors’ response: We appreciate your wonderful view, and we've accepted and incorporated it in the revised version (line number 316 and table 3 page 18).

Editor: Logic of clinical application:

It is suggested that the author invite a clinical practitioner working with the investigated population to go through the study. The author highlights the correlation between depression-anxiety-somatic symptoms as the main outcome; alcohol use, sex and more are correlated with unfavorable mental health situation. However, these findings did not provide novelty to the field. The value of this study is the unique study population. The author could have better elaborate how the results could be explained by environmental factors specific in Northwest Ethiopia and how that could affect policy making/clinical practice.

Authors’ response: We acknowledged your advice, and in the amended paper, we attempted to reflect the particular destiny of adolescents in Northwest Ethiopia (line number 106-107 and 461-462).

Editor: Writing quality:

Please correct all the typos, lower case/upper case, repetitive content, inconsistent font, content generated during the reversing process which supposed to be deleted from the formal manuscript. These are the red flags for the editor that suggested lacking effort in proofreading

Authors’ response: Thank you for your intelligent idea; based on it, we proofread the paper several times and attempted to eradicate all mistakes, upper and lower case differences, and inconsistencies.

Editor: Language:

The syntax is confusing. Please consider having a senior author or peer to help with better expression.

Authors’ response: Thank you for your comments; we consulted senior authors and attempted to adjust the syntax based on their suggestions.

Reviewer's Responses to Questions

Comments to the Author

1. Is the manuscript technically sound, and do the data support the conclusions?

Reviewer #1: Yes

Reviewer #2: Yes

2. Has the statistical analysis been performed appropriately and rigorously?

Reviewer #1: Yes

Reviewer #2: Yes

3. Have the authors made all data underlying the findings in their manuscript fully available?

Reviewer #1: Yes

Reviewer #2: Yes

4. Is the manuscript presented in an intelligible fashion and written in standard English?

Reviewer #1: Yes

Reviewer #2: Yes

5. Review Comments to the Author\\

 

2. Response to Reviewer #1: 

Authors’ response: First of all, we would like to say thank you for your great effort, commitment, humility, and time to comment our manuscript. Next, we would like to respond to the comments point-by-point. All the points raised are corrected accordingly in the main manuscript and put in track change.

General Feedback!

Reviewer 1: The write up, grammar and integrity is good

Eligibilitycriteria

Reviewer #1: Was there no exclusion criteria used?

Author response: We appreciate the reviewer's suggestion, night preparatory and high schools students, as well as those who moved in from other areas after January 8, 2022, were excluded; this was already covered in the initial submission's lines 155, 156, and 157. We've also included it from line 137 to 139 in the updated version..

Operational Definition

Reviewer #1: Do you consider the person with PHQ-9A score of less than 9 have no depression? A person with a PHQ-9A score of 5-9 has mild depression. Do you think missing these segments of the population has effect on your result?

Author response: Thank you for the input. We used the cut-off point above 9, as originally specified in line 217 to 218 of the initial submission. We used the cut-off point above 9, as described in lines 217 to 21 of the first submission. We were also concerned about missing people with mild depression, but we made an effort to read many papers on the threshold for diagnosing depression in the general population. Even though the cut-off point used by clinicians and researchers varies depending on the clinical group, a PHQ-9 score of 10 or higher is more consistently and strongly recommended for community depression diagnosis (Gilbody, Richards et al. 2007), since it is consistent with DSM V criteria for depression diagnosis.. 

Strength and Limitation of the study

Reviewer #1: ‘Assed’: Good to write in full assessed on line 955

Author response: Thank you, we have accepted and modified it (Line number 543).

3. Response to Reviewer #2: 

Authors’ response: First of all, we would like to say thank you for your great effort, commitment, humility, and time to comment on our manuscript. Next, we would like to respond to the comments point-by-point. All the points raised are corrected accordingly in the main manuscript and put in track change.

Reviewer #2: To the best of my knowledge the study was well executed, the data sound and the analyses appropriate (though I leave the final judgement on analysis to reviewers more expert than me in SEM). The manuscript is on an important topic, it is reasonably well-written and the study findings ought to be brought to light. 

However, considerable re-organization and making the paper far more concise will be required to meet the standards for publication, in any journal.

Author response: Thank you for your insightful suggestion. We modified and corrected the document in the revised manuscript according to your general and specific comments putted in the attached document as follow:

Reviewer #2: General comments: 

It would be much easier to read and review if you used the usual conventions for manuscript preparation e.g. the use subtitles/sections such as Introduction (which does not require a subtitle) Methods (with subsections Study Design and Context, Participants [including all details regarding sampling], Measures/Instruments (choose one and see note below about how to organize), Results (with subsection “Plan for Analysis” then report the actual analysis and results) and finally Discussion. Please see for example, the APA version 7 for how to prepare the document (although you would want to use numeric referencing as required by PLoSONE).

The following link may be helpful.

https://www.apa.org/pubs/journals/resources/manuscript-submission-guidelines

Using the conventional section for a manuscript will help to avoid redundancy or making reference to things not yet described, which occurs fairly frequently in the paper. 

Authors Response: Thank you for your thoughtful suggestions. We amended our article in response to your comments and in the revised paper; we followed the standard manuscript writing conventions: introduction, methods and materials (research design and context, participants, measurement), results, discussion and conclusion (IMRDC).

Reviewer #2: Please also note that indenting each new paragraph makes the manuscript much easier to read, especially to review.

Author Response: Thank you. We accepted your comments and modified them in the revised manuscript.

Reviewer #2: Another main consideration here is the paper organization – this applies to all aspects of the paper (though much less so the introduction, which is sensibly written) but even more to the methods, results and discussion. The reporting of measures, results and discussion are far to piece-meal and list-like. It makes it very hard for the reader to digest or to grasp the big picture of the findings, let alone to make sense of them.

Author Response: We corrected the above comments in the updated article and tried to organize the methodology, results, and discussion section in accordance with the precise remarks comments provided for us.

Reviewer #2: I understand this is primarily an exploratory paper (no apparent hypotheses are being tested) but suggest the authors consider 2-3 primary predictors and then perhaps others that are secondary. Report the results (and also address in the discussion) the primary predictors first looking at impact on all outcomes (depression, anxiety and somatic illness) simultaneously, not in different paragraphs. Then summarize these findings (in the discussion, discuss them). Then concisely report the picture for the secondary predictors, also looking at impact on all outcomes simultaneously, followed by a brief summary. Readers can consult the tables for the different outcomes if they want the specific findings for each predictor and outcome combination. 

Author Response: We accept the provided suggestions and corrected them in the revised manuscript (example: line number 464-465).

Reviewer #2: In other words, the results and discussion need to be better and more tightly organized. The authors need to consider what are the most important findings to share, to paint the big picture and how it fits into the rest of the literature, and then organize the paper to highlight these findings. 

Authors’ response: your comment is accepted and we tried to refine the most important findings and tried to compare with the existed body of knowledge and literature (example: page 23 from the result part and page 29 from discussion part).

Reviewer #2: Specific recommendations: 

Title:

Depression, Anxiety, Somatic symptom and their determinants among High School

and Preparatory School Adolescents in Gondar Town, Northwest Ethiopia: Non-recursive

Structural Equation Modeling. 

Suggested Revised Title:

Determinants of adolescent depression, anxiety, somatic symptoms in Northwest Ethiopia. Authors’ response: Thank you, we have corrected the title as per the recommendation (line number 1-3). 

Abstract 

Reviewer #2: In general (throughout the paper), care should be taken to guard the anonymity and confidentiality of the participating sample. See e.g. the suggested revised title. In the abstract, 2ndparagraph “…to 1407 adolescents in northwest Ethiopia”.

Authors’ response: We took your excellent advice. We modified it in the revised manuscript (example: line 20).

Reviewer #2: Please remove all names of particular schools and keep description to the region where the study was conducted, the two different types of schools and how they differ and keep sample demographics and so on in Participants section. Comment to remove the names of particular schools applies throughout the paper. 

Authors’ response: thank you for your insightful suggestion, based on your advice, the names of the schools have been deleted from the entire version of the amended manuscript in order to maintain the confidentiality of the participants.

Results section in the abstract:

Reviewer #2: It would be better to organize and to make more concise the results reported. Perhaps according to outcomes? Determinants of anxiety, then of depression, then interrelated findings for these variables.

Authors’ response: We appreciate the reviewer's suggestions to improve the quality of our article. The comment is accepted and modified (line number 25 to 30).

Reviewer #2: Introduction (Does not need to be labeled “Introduction”)

Authors’ response: We appreciate your suggestion. The introduction is listed as one of the publication's components in the submission rules for the PLOSONE journal. We corrected it according to the journal’s manuscript preparation guideline (line 41).

Reviewer #2: At line 68, please place a comma after “…adolescents, is characterized…”

Authors’ response: Thank you for the comment, we accepted the comment and corrected it in the revised manuscript (line number 58).

Reviewer #2: Lines 70-71 Suggest move the sentence on the impact of depression combined with anxiety to the end of the next para on anxiety.

Authors’ response: Thank you for your comment; we have accepted and modified it (line number 80-81).

Reviewer #2: Line 76, I think you mean “threats” and not “treats”

Authors’ response: Yes, it to mean threats, and we corrected it in the revised manuscript (line 71).

Reviewer #2: Line 90 “..think or experience feelings about symptoms…” 

Authors’ response: Thank you for the suggestion and we have corrected it in the revised manuscript (line number 88).

Reviewer #2: Line 101-102, the sentence is not complete. Please attach sentences starting at 101 and 104 to the para above.

Authors’ response: We acknowledged the suggestion and we corrected it in the revised manuscript (line number 64, 81 and 95).

Reviewer #2: Line 104-106 What is the nature of the relation? Positive or negative?

Authors’ response: Thank you for the comment. We have accepted and modified it (line 98).

Reviewer #2: Line 113-114 Suggest, “Finally, individuals with… may experience physical harm from unnecessary medical procedures”

Authors’ response: thank you for your suggestion, we agreed with your suggestion and we corrected it in the revised manuscript (line number 94 -95).

Reviewer #2: Line 135 Not clear here what you mean by “as an input toward sustainable development”.

Authors’ response: Thank you for your question, an input towards the sustainable development goal is to say; the slogan of the SDG is "no health without mental health," and in the SDG focusing on adolescents, mental health is the main target, because more than half of mental health disorders begin in adolescence and progress to adulthood. SDG agenda includes research as one of the initiatives to address the mental health problem. In light of this, our study will help to accomplish the Sustainable Development Goal. That is why you say it is a contribution to sustainable development.

Methods (Include this major subtitle)

Reviewer #2: Line 138 Suggest “Study Design and Context” as first subsection.

Authors’ response: Thank you for the suggestion, we accepted your suggestion and it is modified in the revised manuscript (line number 123).

Reviewer #2: Sample Description – Label as Participants

Authors’ response: Thank you for the suggestion, we accepted the suggestion and it is corrected (line number 135).

Reviewer #2: Please justify the use of this particular population of Ethiopian adolescents. How might this contribute to or constrain generalization of study findings? Is this a convenience sample?

Authors’ response: Thank you for the comments and questions, based on your suggestion we included an idea related with the use of particular population of Ethiopian adolescents in the revised manuscript from line number 129 to 130 and the sampling is stratified random sampling rather than convenient sampling.

Reviewer #2: Line 149, suggest remove the section on population as this information is given elsewhere.

Authors’ response: we accepted the suggestion and population section was omitted in the revised manuscript.

Reviewer #2: Lines 162-169 please correct the use of parentheses, 1 set with commas between different types of free parameters). Currently, very hard to follow.

Authors’ response: thank you for the comment; we simplified it in the revised manuscript (line number 140-143).

Reviewer #2: Line 171, if the sampling was stratified by the 2 different types of schools, say so here or above under context. If stratified first by type of school and then grade, at this point please include “(see below)” otherwise the comment here about a 2-stage sampling procedure seems out of place. 

Authors’ response: Thank you for the suggestion, we accepted your idea and in the revised manuscript we have corrected it (line 162).

Reviewer #2: Line 183, under sampling procedure, I would not name the particular schools involved in the study as this poses a risk to participant confidentiality. Knowing the names of particular schools does not enhance the study reporting. 

Authors’ response: We acknowledge the comment of the reviewer, in the rewritten version as a whole; names of specific schools are omitted.

Reviewer #2: Line 201, please place these variables in a table, and move them to supplementary text. Place a footnote or in the text refer the reader to the table for a summary of the key variables in the study. 

Authors’ response: We accepted the suggestion and modified it according to the feedback(line number 170).

Reviewer #2: Line 216, suggest remove the section on “Operational definitions” and instead place the relevant information in the next section, immediately after you describe the measurement instruments. So after describing the depression measure very briefly, then say the criteria you used for categorization in the study. 

Authors’ response: Thank you for the insightful suggestion, the section operational definition was omitted in the revised manuscript and information’s in the operational definition is incorporated in measurement section (page 10).

Reviewer #2: Line 245 Suggested “self-reported questionnaire”. 

Authors’ response: Your suggestion is acknowledged and we have corrected it (line number 173).

Reviewer #2: Please be sure that in the sample description you indicate the percentage/proportion/number of sighted and sight challenged individuals and proportion by school type and gender perhaps? Sight capabilities of participants should not suddenly appear here.

Authors’ response: We thank the reviewer for the comment; we accepted and incorporated it (line number 173-174).

Reviewer #2: Clarify if the questionnaires were administered in paper and pencil format or on a computer or other device. 

Authors’ response: thank you for the comment, we accepted your comment; it was administered in pencil and paper format and it is included in the revised manuscript (line number 172).

Reviewer #2: 249-251, I do not understand the use of “The tool for this study here”. Do you mean “Questionnaires were designed to gather data on…” (insert your list of domains here).

Authors’ response: Thank you for the suggestion. yes it is to mean, questionnaire were designed to gather data on socio-demographic characteristics, behavioural factors......and we have corrected it in the revised manuscript (line number 176-178).

Reviewer #2: Suggest place all of your measures in a table with columns for 1. Instrument name and citation 2.Brief description and 3. Criteria for categorization you included above. 

Authors’ response: We accepted your suggestion and in the revised manuscript all of the measures are placed in the table (table 1, page 9-10).

Reviewer #2: Line 268, there should be a subheading for results. 

Authors’ response: Thank you for your feedback, we accepted the comment and corrected it in line number 181 of the revised manuscript.

Reviewer #2: Line 271, does the translation apply to all of the questionnaires (were they arranged into one long document?)

Authors’ response: thank you for the question, yes translation was applied for the entire questionnaire after we arrange it in subheading.

Reviewer #2: Although the Data Quality and Management efforts are laudable, this section is long and superfluous here. 

Authors’ response: In the revised manuscript, after we put very essential parts in the main document, the remaining data quality assurance parts were moved to supplementary file (line 194).

Reviewer #2: Suggest, remove lines 269 to first part of 271, you already mentioned trained data collectors above.

Authors’ response: We accepted the wise advice you provided, in the revised manuscript we have corrected it (line 182).

Reviewer #2: Great that you checked face validity, construct validity and reliability. For the latter to please indicate very briefly how you checked these.

Authors’ response: Thank you for your suggestions to improve our manuscript, the construct validity was checked by using Average variance extracted and reliability was checked by using Cronbach’s alpha (line number 186-187 and 191-192).

Reviewer #2: On second thought; perhaps you could have a “Measures” section, where you include the table suggested above. Before the table, include all information related to the measures such as translation and also efforts to get at validity and reliability. Include a footnote there and describe the pilot study briefly in the footnote. What was done? What did you find? Or place the pilot study in supplementary materials and refer to it in the paper.

Authors’ response: Thank you for the suggestion; based up on your idea the detailed part of the pilot study was placed in the supplementary material (S3 annex 1) (line 194).

Reviewer #2: Note, it is not clear in what sense the pilot study constituted “participatory” research. Needs to be clarified wherever you put the pilot study description.

Authors’ response: Thank you, according to our search, there are two categories of pilot studies: participatory and non-participatory. So a participatory pilot study is a form pilot study in which participants are aware that they are taking part in a preliminary study and are invited to provide feedback to improve the questionnaire.

Reviewer #2: Lines 279-286 Suggest remove the rest of this paragraph. What does it mean to say “After data entry, data recoding, and missing value management were considered accordingly.” If important, it could go at the start of the Results section, perhaps right before “Plan for Analysis”.

Authors’ response: Thank you, in the revised manuscript that unnecessary information’s from line number 279 to 286 are omitted.

Reviewer #2: Are the Results given on page 14 for the pilot study? If yes, suggest the entire pilot study goes in Supplementary Material. Simply say in the manuscript that you did a pilot study to further investigate the validity of the questionnaire. Then refer the reader to Supplementary Material.

Authors’ response: We express gratitude for the reviewer's suggestions to improve our manuscript; yes it was for the pilot study and in the revised manuscript the detail part the pilot study was placed in the supplementary material (line number 194).

Reviewer #2: The pilot study takes up a lot of space in the manuscript but, if I understand correctly, it is meant to convey background information to support the use of the questionnaire in the main study. The manuscript body should focus on the main study.

Authors’ response: Thank you for your suggestion, sure the manuscript body should focus on the main study and we tried to minimize pilot study part and we moved majority of pilot findings to supplementary file.

Reviewer #2: Remove the sentence end of line 320 to 321. Connect the sentence at line 34 to the previous paragraph. 

Authors’ response: Accepted and modified (line number 201).

Results?

Reviewer #2: The results section is generally far too long, with too many explanatory details about how structural equation modeling works. If you aim to show your thoroughness in carefully undertaking this analysis, only mention the specific ways in which you did this in the paper. Assume the audience knows something about SEM. 

Authors’ response: We acknowledged your recommendation, and in the updated manuscript, we attempted to minimize information about how the model works.

Reviewer #2: Suggest re-ordering reporting of results to follow same order as mentioned in introduction and as the order of measures described in the methods section – depression, anxiety, somatic symptoms, then interrelated of these.

Authors’ response: Thank you for your suggestions to improve our manuscript, we accepted the suggestion and in the revised manuscript we follow the same pattern as the introduction; depression, anxiety and somatic symptom. (Example: see line 331, 337 and 343 for the prevalence of depression, anxiety and somatic symptom respectively). 

Reviewer #2: It is difficult the follow the list of findings in the paragraphs for each of these outcomes. Could you organize into direct effects only, unpack these, then indirect effects, unpack these and then perhaps deal with variables with both direct and indirect effects on the outcome that is the focus of that paragraph (depression, anxiety or somatic symptoms).

Authors’ response: Thank you, we accepted and modified it according to your feedback; (page 23-26).

Reviewer #2: Line 677. “…Bangladesh use a cutoff above five…” please add “whereas in the current study, we used a cutoff on the X measure of Y”. Fill in X and Y accordingly.

Authors’ response: Thank you for the comment; we accepted it. However, based editors’ suggestion specific details about previous studies were not included in the revised manuscript.

Reviewer #2: Interesting point. Please elaborate briefly on how socio-cultural factors might help to account for discrepancies in findings here. 

Authors’ response: We acknowledge the reviewer comments to improve the quality of our manuscript, in the revised manuscript we tried to elaborate how socio cultural factors may account for the discrepancy in mental health illness (line number 455 to 461).

Reviewer #2: Line 683, It could also be that females are more likely to report on their depression! 

Cultures vary in terms of how males and females ought to deal with things such as depression and this might have an impact on how willing or unwilling males and females are report being depressed or having other mental health challenges. 

Authors’ response: Thank you for the suggestion, we accepted your suggestion and it is included in the revised manuscript (line number 473-476).

Reviewer #2: Line 689, regarding the following sentence:

“Besides, adolescent girls are more likely to have negative life events in relation to their parents and peers, and females are more emotion focused and have a distracting coping style than males”

Please unpack these findings so the reviewer can understand exactly what the authors of the cited paper found. It is not clear to me why adolescent girls are more likely to have negative life events in relation to their parents and peers. What does it mean to be “more emotion focused and to have a distracting coping style than males”?

Authors’ response: We respect your thoughtful observation, after we dig out different source, there is no tangible evidence which help us to elaborate this idea and in the revised manuscript the justification adolescent girls are more likely to have negative life events in relation to their parents and peers, and females are more emotion focused and have a distracting coping style than males are omitted 

Reviewer #2: Line 699 – 700, these claims need supporting references.

Authors’ response: Accepted and cited (line number 513).

Reviewer #2: Line 701, it is also possible that people who are more depressed are more likely to use alcohol as a coping mechanism. 

Authors’ response: We appreciate your suggestion and agree with your worry. People who are depressed are more inclined to use alcohol as a coping strategy. It is difficult to tell whether alcohol usage or depression develops first, and this is one of our study's shortcomings (line number 546-547).

Reviewer #2: Line 707. It could also be that being depressed means you have low motivation to study or even that you do not see any point in striving in school given low or no expectations regarding a well-adjusted, productive or successful future. 

Authors’ response: Thank you for your thoughtful comment. As with the link between drinking and depression, bad academic performance may be caused by depression, and the two conditions may be correlated in both directions. However, because of the cross-sectional nature of the data, we are unable to determine which comes first in our study and you failed to consider how depression affects academic performance. We included it to the study's limitations in order to concern future researchers (line number 547-547).

Reviewer #2: In reporting results, it would be group according to outcome e.g. depression. Then review of the variables with a direct effect, briefly unpack, those with indirect effects, briefly unpack and then with both direct and indirect impact, then describe – in other words follow the same list and organization suggested for reporting results. 

Authors’ response: We accepted the wise advice you provided, and in the revised manuscript we unpack the direct effect first, then indirect effect and finally both direct and indirect effect were unpacked (page 23-26).

Reviewer #2: Losing a loved one is a traumatic and by definition a sad experience. This is not a surprising finding and does not seem to require much more unpacking. Respondents could still be recovering from this loss.

Authors’ response: Even though the depressive symptoms persist in some individuals with a history of death loved one, majority of the individuals may recover from it by themselves. So we accepted your suggestion and the justification related with death of loved one is omitted in the revised manuscript.

Reviewer #2: Line 743 and 744, interesting finding re: private school and higher rates of depression but in the results you report both direct (negative) and indirect (positive) with a small net positive effect. It’s worth referring to this and elaborating briefly here.

Authors’ response: We respect your thoughtful observation, being in private increase the likelihood of depression both directly and indirectly with a small net positive effect. Even though the net effect is little, it is still important; although it is important to prioritize and elaborate on aspects that have a greater impact, we still believe that it is important to examine or explore small but important significant effects.

Reviewer #2: Line 777-778, if you are going to suggest that cultural differences might account for inconsistencies in results across studies, good to give an example of exactly how this might play out.

Authors’ response: We appreciate the reviewer's efforts to enhance the quality of our research; the updated manuscript describes the role of cultural variation to the inconsistencies (line number 455-461 and 474 -476). 

Reviewer #2: Lines 786-788, “…students with high self-rated academic ability will be more

likely to make decisions, express their feelings without feeling embarrassed” Doing better academically is likely to increase confidence compared to doing poorly and these might lead to more confidence in decision making but it is not clear to me why this would increase the willingness to express feelings without being embarrassed and necessarily to more optimism.

Authors’ response: We appreciate your query and your agreement that adolescents who excel academically will be more valued by their family, friends, and teachers, which will help them gain the confidence to communicate their feelings. It is essential for young people's successful social development. Students who do well in school have a lower risk of suicide, which makes them more prepared to transition into adulthood. Additionally, adolescent who have excelled academically will say, "I can do anything," and those around them will support and encourage them with resources and ideas, which will improve their decision-making abilities.

Reviewer #2: Lines 799-805, again it could be that women are more likely to disclose struggles with anxiety compared to men.

Authors’ response: Accepted and incorporated (line number 473-476).

Reviewer #2: The point above about organizing study findings in the results and discussion applies to all of the outcomes. Another way to organize might be to treat each predictor in turn and then consider the impact on all three outcomes. There were some common findings for predictors with different outcomes e.g. alcohol consumption and private school both had a positive relationship with depression and also with anxiety. The explanations for the impact on different outcomes might be the same, so would only need to be mentioned once.

Authors’ response: We took your wise advice into account. As we already indicated for previous comment, we attempted to break down the direct effect first, followed by the indirect effect, and then both the direct and indirect effects in the result part. In the discussion section, we first talk about the factors that affected all of the outcomes, and then we talk about the factors that had a particular impact on one of the outcome (page 23-25 for result part and page 27-31 for discussion part).

Reviewer #2: Study Strengths and Limitations

Generalizability is also constrained by the specific, targeted sample. The authors should recognize and address this point. Is the sample nonetheless diverse and representative of Ethiopian adolescents and if yes, how do you know this and what then does this mean for study generalizability but also for policy and practice in Education, in Ethiopian society?

Authors’ response: Your thoughtful suggestion was accepted, and it has been incorporated in to the limitation part (line number 543 to 544).

Reviewer #2: In several places in the discussion, the authors raise the point of cultural differences and how these might account for discrepancies in study findings across studies conducted in different cultures. But they never really expand on the nature of those differences. Yet this would be so interesting here and contribute to the literature on factors that contribute to mental health outcomes and their context specificity or universality, which has knock on effects for how to mitigate these struggles or how to approach treatment. 

Authors’ response: Acknowledged, and we provide a justification (line number 455-461 and 474 -476).

Reviewer #2: It would be better if the recommendations were not stated in bullet form.

Authors’ response: Thank you, we have accepted and modified it in the revised manuscript (line number 555 -562).

Line 987 and on

Reviewer #2: To avoid the possibility of stigma surrounding coping with and addressing mental health issues, preventative and coping strategies at every level might be better if offered to everyone, not targeting those with certain characteristics highlighted by the study findings. Targeted treatment might invoke stigma which could actually negatively impact mental health. 

Authors’ response: Thank you accepted and updated in the revised manuscript (line number 556 to 557).

Reviewer #2: Line 991, see suggestion above

Authors’ response: Accepted and modified (line number 557-558).

Reviewer #2: Line 999, I do not think you established that drinking alcohol caused mental health issues. If this is a coping mechanism or addiction is in play, it might be better to seek treatment for the mental health issues as alcohol consumption is likely to come up as part of this. Parental sudden restriction but exacerbates symptoms without expert treatment. 

Authors’ response: Thank you for your comments. Due to the cross-sectional nature of the data, we did not establish a causal relationship between alcohol intake and mental illness and we have putted it in the limitation (line number 547-548).

Reviewer #2: Line 1001, again alcohol consumption might be a coping mechanism or indeed a symptom of struggles rather than causally related. Not as well that you only asked about “ever consumption” not about how much or how frequently or for how many years consumed (another study limitation).

Authors’ response: Accepted, and we incorporate it in the limitation (line number 546-547).

Reviewer #2: Line 1006, interesting suggestion regarding out-of-school adolescents. It would be good to provide prevalence data – these might be indicators of struggles with mental health in themselves. 

Authors’ response: Thank you for your suggestion; since we did not collect data on out-of-school adolescents, we are unable to give prevalence information at this time. However, we will conduct another study on them in the future.

---

## [Decision Letter · Decision Letter 2]

4 Sep 2023

PONE-D-23-02244R2Determinants of adolescent’s depression, anxiety, and somatic symptoms in Northwest Ethiopia: A Non-recursive structural equation modeling.PLOS ONE

Dear Dr. Gebreeqziabher,

Thank you for submitting your manuscript to PLOS ONE. After careful consideration, we feel that it has merit but does not fully meet PLOS ONE’s publication criteria as it currently stands. Therefore, we invite you to submit a revised version of the manuscript that addresses the points raised during the review process.

We look forward to receiving your revised manuscript.

Kind regards,

Ching-Fang Sun, MD

Academic Editor

PLOS ONE

Journal Requirements:

**Additional Editor Comments (if provided):**

Please address reviewer's comment as advised.

**Reviewers' comments:**

Reviewer's Responses to Questions

**Comments to the Author**

1. If the authors have adequately addressed your comments raised in a previous round of review and you feel that this manuscript is now acceptable for publication, you may indicate that here to bypass the “Comments to the Author” section, enter your conflict of interest statement in the “Confidential to Editor” section, and submit your "Accept" recommendation.

Reviewer #1: All comments have been addressed

Reviewer #2: All comments have been addressed

2. Is the manuscript technically sound, and do the data support the conclusions?

Reviewer #1: Yes

Reviewer #2: Yes

3. Has the statistical analysis been performed appropriately and rigorously? 

Reviewer #1: Yes

Reviewer #2: Yes

4. Have the authors made all data underlying the findings in their manuscript fully available?

Reviewer #1: Yes

Reviewer #2: Yes

5. Is the manuscript presented in an intelligible fashion and written in standard English?

Reviewer #1: Yes

Reviewer #2: No

6. Review Comments to the Author

Reviewer #1: (No Response)

Reviewer #2: In general the paper is greatly strengthened and I appreciate the close attention to the reviewers' comments and the changes made to the manuscript. My main remaining comments are 1. As another reviewer/editor has pointed out, the occurrence of anxiety, depression and somatic symptoms in adolescence is not a new hypothesis or finding. The unique contribution of the paper is documenting these issues in an Ethiopian adolescents and the exploration of potential contributing factors in that population. Hence, I would start the paper by making this point before moving on to define anxiety, depression (I would use this order) and somatic disorders, their prevalence in other regions, other parts of Africa and then Ethiopia, perhaps pointing out the dearth of research on Ethiopian adolescents in the conclusion of this section of the introduction. and 2. Efforts at writing in English which may not be the native language of the authors is alway admirable, in my view, and should not prohibit the publishing of important and interesting data as we have here. However, for maximum impact, it would make sense for the authors to have a reviewer read the paper for smooth flow in the English language as a final polish before submitting. In general, the paper is clear but this is scientific writing and precision in language counts. No criticism of the authors, just a suggestion to "professionalize" the writing further before being considered for acceptance. I wish the authors the best of luck in getting the paper accepted for publication.

7. PLOS authors have the option to publish the peer review history of their article (what does this mean?). If published, this will include your full peer review and any attached files.

Reviewer #1: No

Reviewer #2: **Yes: **Tracy Solomon, PhD

---

## [Author Response · Author response to Decision Letter 2]

7 Sep 2023

eviewer #2: In general the paper is greatly strengthened and I appreciate the close attention to the reviewers' comments and the changes made to the manuscript. My main remaining comments are 1. As another reviewer/editor has pointed out, the occurrence of anxiety, depression and somatic symptoms in adolescence is not a new hypothesis or finding. The unique contribution of the paper is documenting these issues in an Ethiopian adolescents and the exploration of potential contributing factors in that population. Hence, I would start the paper by making this point before moving on to define anxiety, depression (I would use this order) and somatic disorders, their prevalence in other regions, other parts of Africa and then Ethiopia, perhaps pointing out the dearth of research on Ethiopian adolescents in the conclusion of this section of the introduction. and 2. Efforts at writing in English which may not be the native language of the authors is alway admirable, in my view, and should not prohibit the publishing of important and interesting data as we have here. However, for maximum impact, it would make sense for the authors to have a reviewer read the paper for smooth flow in the English language as a final polish before submitting. In general, the paper is clear but this is scientific writing and precision in language counts. No criticism of the authors, just a suggestion to "professionalize" the writing further before being considered for acceptance. I wish the authors the best of luck in getting the paper accepted for publication.

Author:We tried to address the comments point by point

---

## [Decision Letter · Decision Letter 3]

7 Nov 2023

PONE-D-23-02244R3Determinants of adolescents’ depression, anxiety, and somatic symptoms in Northwest Ethiopia: A Non-recursive structural equation modeling.PLOS ONE

Dear Dr. Gebreegziabher,

Thank you for submitting your manuscript to PLOS ONE. After careful consideration, we feel that it has merit but does not fully meet PLOS ONE’s publication criteria as it currently stands. Therefore, we invite you to submit a revised version of the manuscript that addresses the points raised during the review process. Please revise the manuscript as advised by the reviewer. This editor assume this will be the last round of reversion.

We look forward to receiving your revised manuscript.

Kind regards,

Ching-Fang Sun, MD

Academic Editor

PLOS ONE

Journal Requirements:

Reviewers' comments:

Reviewer's Responses to Questions

**Comments to the Author**

1. If the authors have adequately addressed your comments raised in a previous round of review and you feel that this manuscript is now acceptable for publication, you may indicate that here to bypass the “Comments to the Author” section, enter your conflict of interest statement in the “Confidential to Editor” section, and submit your "Accept" recommendation.

Reviewer #1: All comments have been addressed

Reviewer #2: (No Response)

2. Is the manuscript technically sound, and do the data support the conclusions?

Reviewer #1: Yes

Reviewer #2: Yes

3. Has the statistical analysis been performed appropriately and rigorously? 

Reviewer #1: Yes

Reviewer #2: Yes

4. Have the authors made all data underlying the findings in their manuscript fully available?

Reviewer #1: Yes

Reviewer #2: (No Response)

5. Is the manuscript presented in an intelligible fashion and written in standard English?

Reviewer #1: Yes

Reviewer #2: Yes

6. Review Comments to the Author

Reviewer #1: (No Response)

Reviewer #2: This paper is so very much improved from my previous reading of it. The study is well executed, the writing so much improved and the analysis thorough and responsibly reported. I have made suggestions for further tightening of writing only - I sincerely hope to be helpful here not to delay the manuscript processing further. My comments have been made on the manuscript PDF directly. Please see the attachment that bears my initials TS at the end. If the authors attend to these minor suggestions the paper will be in fine shape for publishing in my view.

7. PLOS authors have the option to publish the peer review history of their article (what does this mean?). If published, this will include your full peer review and any attached files.

Reviewer #1: No

Reviewer #2: **Yes: **Tracy Solomon, PhD

---

## [Author Response · Author response to Decision Letter 3]

7 Dec 2023

Reviewer 2

Reviewer #2: This paper is so very much improved from my previous reading of it. The study is well executed, the writing so much improved and the analysis thorough and responsibly reported. I have made suggestions for further tightening of writing only - I sincerely hope to be helpful here not to delay the manuscript processing further. My comments have been made on the manuscript PDF directly. Please see the attachment that bears my initials TS at the end. If the authors attend to these minor suggestions the paper will be in fine shape for publishing in my view.

Authors’ response: Dear Dr. Tracy Solomon, First and foremost, we would like to extend our sincere gratitude for your exceptional effort, dedication, humility, and the significant amount of time you devoted to reviewing and providing comments on our paper. We highly value your feedback, and we would like to address each comment point by point as follow.

1. ABSTRACT 

Reviewer 2: line number 25% missing 

Author response: Thank you for your suggestion. We have made the necessary corrections in the revised manuscript (Kindly look at line number 25).

Reviewer 2: Line number 27: Ever use of alcohol, change it to alcohol use 

Author response: We appreciate your insightful suggestion, and we have addressed it in the revised manuscript. We kindly request you to review line number 27 for the specific changes made.

2. INTRODUCTION

Reviewer 2: line number: which is associated with increased …?

Authors’ Response: Thank you for your valuable comments. We have incorporated them into the revised manuscript, and we kindly request you to review the end of line 42 to 43 for the specific changes made.

Reviewer 2: involving significant 

Authors’ Response: We greatly appreciate your valuable input. Taking your feedback into consideration, we have made the necessary corrections in the revised manuscript (kindly look at line number 44 in the revised manuscript). 

Reviewer 2: Line number 46: This has been linked to…

Author response: Accepted and modified (kindly looks at line 45 in the revised manuscript).

Reviewer 2: Line number 56: add most 

Author response: Thank you for your feedback. We have accepted your suggestions and made the appropriate modifications in the revised manuscript (kindly looks at line number 55 ) Reviewer 2: line 58: As mentioned in the earlier review, why not start with the most common anxiety and then move to the second common depression. Odd start to this paragraph if the next one is about anxiety 

Author response: Thank you for providing your comments once again. We have carefully considered your suggestions and made the necessary amendments in the revised manuscript. We kindly request you to review lines 57-83 for the specific changes implemented.

Reviewer 2: line number 96: Female sex: female 

Author response: Accepted and modified (kindly looks at line 95 in the revised manuscript).

Reviewer 2: line number 97: Extra school change to extra curricular

Author response: Thank you for providing your feedback. We have carefully corrected it in the revised manuscript (line number 95).

Reviewer 2: Line 99: remove although and add yet after the comma

Author response: thank you, accepted and modified (kindly looks at line number 98-99 in the revised manuscript)

Reviewer 2: Line 103: although adolescents 

Author response: thank you for your feedback; we have corrected it according to your suggestion (kindly looks at line number 102).

Reviewer 2: Line 108: replace moreover..And make it the third sentence 

Author response: We greatly appreciate your suggestion. It has been accepted, and we have made the necessary modifications in the revised manuscript accordingly (kindly look line in the 108).

Reviewer 2: Adolescents in northwest Ethiopia are especially vulnerable to mental illness, due to repeated internal conflicts in northern Ethiopia: Suggest making this your second sentence in paragraph.

Author response: Thank you for your input. We have accepted your suggestions and made the appropriate modifications (line 107-108). 

Reviewer 2: Line 116: Therefore, this study will assess the prevalence and determinants of … and the relationship between these variables.

Author response: accepted and corrected (line 115).

3. METHODS AND MATERIALS 

Participants: 

Reviewer 2: line 141suggests gives total sample size here, put information regarding sample size calculation in footnote or in supplementary material. Then give description of the sample, perhaps in summary table. Proportion private and public schools, identified as male and female, by grade include average age with range or sd, any other demographic info if available such as SES, sighted vs blind individuals 

Author response: Thank you for providing your invaluable comments. We have accepted and incorporated your suggestions into the revised version (kindly looks at page 7-9).

Reviewer 2: Line 159: this paragraph fine to include in the sample 

Author response: thank you, we have accepted your feedback and made the necessary modifications (line number 140-141).

Reviewer 2: Line 169: I don’t know that you need both figure and textual description here.

Author response: Thank you for your suggestion and feedback. As per your advice, we have decided to omit the textual description (kindly looks at line 141) from the revised version.

Data collection procedures and tools 

Reviewer 2: Line 178: did participants complete the questionnaire in group setting or were they assessed individually. Where were they assessed? In the school library? Classroom? Quite area in the school? How long was the test session? Were breaks taken as needed? 

Author response: Thank you for your feedback, which allowed us to incorporate this information. In the revised manuscript, we have included the details that the participants completed the questionnaire in classroom setting, and the completion time varied between 15 and 25 minutes (kindly look line 175-176).

Reviewer 2: Page 10, table 1: ever use of alcohol: please change to alcohol use? 

Author response: We appreciate your suggestion, and we have taken it into account in the revised manuscript. The term "alcohol use" has been included consistently throughout the entire document, as you recommended.

 Data quality assurance

Reviewer 2: Line 185: suggest place this section in supplementary material 

Author response: Thank you for your input. In the revised manuscript, we have moved this section to the supplementary material as per your suggestion (kindly look at line 187).

Ethical consideration 

Reviewer 2: Line 311: as judged by …? State if they were excluded from the study. What was the resulting effective sample?

Author response: Dear Dr. Tracy Solomon, We would like to express our gratitude for bringing this issue to our attention. In our study, non-volunteer individuals were indeed excluded (line number 314-315 in the revised manuscript). However, we did not exclude individuals with moderate to severe depression, anxiety, and somatic symptoms, as their status became known only after they completed the questionnaire.

4. RESULTS 

Reviewer 2: Line number 318: suggest place all of this section under participants, at the start of method section.

Author response: Thank you for your suggestion. We have taken it into consideration and made the necessary modifications accordingly (page 7-9).

Reviewer 2: Line 327: I would also include under sample description-text only. Move the table here to supplementary materials 

Author response: Thank you for your suggestion. We have accepted it and made the necessary modifications in the revised manuscript. Kindly refer to lines 150-155 in the revised version for the specific changes implemented.

Reviewer 2: Line 336: suggest for this section:

“Results “The subtitle:”plan for analysis” then subtitles for each of your study goal “prevalence” “contributing factors “you could combine the last 2 sub-sections. If you will speak to them simultaneously.

Author response: Dear Dr. Tracy Solomon, Thank you for your suggestion. We appreciate your recommendation and acknowledge that including the plan of analysis in the result section is one of the possible options. However, in health science writing, it is also common to present the plan for analysis in the method section, which is the approach we have chosen to follow in this study. In the previous manuscript, we had already incorporated the plan for analysis under the section titled "Data Processing, Model Building, and Analysis" (line 188). Therefore, we did not specifically include the plan of analysis in the result section. Instead, we opted to start the result section with the title "Results," followed by subsections that cover the prevalence of depression, anxiety, somatic symptoms, and contributing factors.

Reviewer 2: Line 356: best under plan for analysis. Condense this section as much as possible, describe in sufficient detail for the reader to understand what you did but micro stuff in supplementary material section. Such a good paper, so thoroughly in approach but this gets lost in overwhelming the reader with all of details in the main manuscript. You don’t have to scarify by cutting all your work but consider moving to supplementary material.

Author response: We appreciate your suggestion, and as per your recommendation, we have incorporated this section into the subsection titled "Data Processing, Model Building, and Analysis." Thank you for your valuable input, which has helped us improve the organization of the manuscript (kindly look line 289-295).

DISCUSSION 

Reviewer 2: Line number 439: we were also interested in the inter-relation between these factors 

Author response: accepted and modified (kindly look line 410 in the revised manuscript).

Reviewer 2: Line 440: “moderate” leave details to the next paragraph 

Author response: Thank you for your valuable comment, which has greatly contributed to strengthening our paper. We have carefully considered your feedback and made the necessary modifications accordingly (line 411).

Reviewer 2: Line 460: do you mean only somatic disorders here or disparity for depression, anxiety to?

Author response: Thank you for bringing this issue to our attention. We intended to highlight the variation in depression, anxiety, and somatic symptoms, and we have now incorporated this information into the revised document (line 430).

Reviewer 2: Line 468: ongoing internal conflict 

Author response: and made the necessary modifications in the revised manuscript. Kindly refer to line 439 in the revised version to see the specific changes that have been implemented.

Reviewer 2: Line 470: suggest you discuss findings with respect to predictors as follow:

1. Higher incidence in female and why?

2. Group academic ability, private school and perceived social support together. For example have an opening statement that all of these had an impact on all 3 of your outcomes , then unpack for each of these, the stress of being in private school with high academic ability….

Authors’ response: Dear Tracy Solomon, We would like to express our sincere appreciation for your valuable efforts in improving our paper. Your feedback has been invaluable, and we have accepted your comment and made the necessary modifications accordingly (kindly looks at page 27 -29).

Reviewer 2: line 483: good here to also mention gender roles, if there are greater disparities in male and female expectation, and these do not favor female, they would have more reason to experience depression, anxiety and possibly somatic disorders.

Author response: Thank you for bringing this issue to our attention. We have accepted your feedback and incorporated the necessary changes in the revised manuscript. Please refer to lines 483-486 in the revised version to review the specific modifications that have been made.

Reviewer 2: Line 484: Please use "alcohol use" throughout your paper

Author response: Thank you for your comment. We have taken your suggestion into account and used the term "alcohol use" consistently throughout the entire revised manuscript. 

Reviewer 2: Line 494: could alcohol be the symptom of these disorders? 

Author response: Thank you for guiding us in this direction. We acknowledge that your suggestion could indeed be beneficial, and we have included it in the revised manuscript (kindly loot line 537-538). 

Reviewer 2: Line 539: subtitle here for relation between variables. Suggest place after the paragraph on prevalence and before the long section predictors. 

Author response: Thank you for your suggestion. We have accepted your suggestion and implemented the required changes accordingly. Please refer to line 445 in the revised manuscript.

5. STRENGTHS AND LIMITATIONS

Reviewer 2: Line 551: public and private schools included and data on under-represented sample in the broader literature are the strengths.

Author response: We have taken your recommendation into account and integrated it into our work (kindly look line 551-552).

Reviewer 2: Line 554: question says “in the last 3 months”

Author response: We have embraced your suggestion and implemented the required edits accordingly (kindly looks at line 556).

---

## [Decision Letter · Decision Letter 4]

17 Jan 2024

Determinants of adolescents’ depression, anxiety, and somatic symptoms in Northwest Ethiopia: A Non-recursive structural equation modeling.

PONE-D-23-02244R4

Dear Dr. Zenebe Abebe Gebreegziabher,

We’re pleased to inform you that your manuscript has been judged scientifically suitable for publication and will be formally accepted for publication once it meets all outstanding technical requirements.

Kind regards,

Ching-Fang Sun, MD

Academic Editor

PLOS ONE

Additional Editor Comments (optional):

Congratulations. It's my pleasure to work with you.

Reviewers' comments:

Reviewer's Responses to Questions

**Comments to the Author**

1. If the authors have adequately addressed your comments raised in a previous round of review and you feel that this manuscript is now acceptable for publication, you may indicate that here to bypass the “Comments to the Author” section, enter your conflict of interest statement in the “Confidential to Editor” section, and submit your "Accept" recommendation.

Reviewer #1: All comments have been addressed

Reviewer #2: All comments have been addressed

2. Is the manuscript technically sound, and do the data support the conclusions?

Reviewer #1: Yes

Reviewer #2: Yes

3. Has the statistical analysis been performed appropriately and rigorously? 

Reviewer #1: Yes

Reviewer #2: Yes

4. Have the authors made all data underlying the findings in their manuscript fully available?

Reviewer #1: Yes

Reviewer #2: Yes

5. Is the manuscript presented in an intelligible fashion and written in standard English?

Reviewer #1: Yes

Reviewer #2: Yes

6. Review Comments to the Author

Reviewer #1: The authors incorporated the feedbacks given and is written well and can proceed to publication. The manuscript is written in standard English and could proceed to publication. The grammar and sentence constriction is acceptable.

Reviewer #2: Congratulations for your perseverance in seeing this manuscript through a lengthy review process. I appreciate your careful attendance to all of my concerns and admire those instances where we disagreed and you held your ground, after thoughtful consideration of my suggestions. The sample on which your research is based is understudied so I will be very excited to see your paper published.

If I may make one tiny final suggestion (I know!) it would be please to change the last word in the title from "modelling" to "model".

7. PLOS authors have the option to publish the peer review history of their article (what does this mean?). If published, this will include your full peer review and any attached files.

Reviewer #1: No

Reviewer #2: **Yes: **Tracy Solomon

---

## [Editor Report · Acceptance letter]

1 Apr 2024

PONE-D-23-02244R4 

PLOS ONE

Dear Dr. Gebreegziabher, 

I'm pleased to inform you that your manuscript has been deemed suitable for publication in PLOS ONE. Congratulations! Your manuscript is now being handed over to our production team.

Kind regards, 

on behalf of

Dr. Ching-Fang Sun 

Academic Editor

PLOS ONE